# CaMKK2 in myeloid cells is a key regulator of the immune-suppressive microenvironment in breast cancer

Luigi Racioppi [1,2], Erik R. Nelson [3,4,5], Wei Huang[1], Debarati Mukherjee[6], Scott A. Lawrence[6,14], William Lento[1], Anna Maria Masci [7], Yiquin Jiao[1], Sunghee Park[6], Brian York[8,9], Yaping Liu[1], Amy E. Baek[3], David H. Drewry [10,15], William J. Zuercher [10,15], Francesca R. Bertani [11], Luca Businaro [11], Joseph Geradts[12,13], Allison Hall[13], Anthony R. Means[8,9], Nelson Chao[1], Ching-yi Chang [6] & Donald P. McDonnell [6]

Tumor-associated myeloid cells regulate tumor growth and metastasis, and their accumulation is a negative prognostic factor for breast cancer. Here we find calcium/calmodulin-dependent kinase kinase (CaMKK2) to be highly expressed within intratumoral myeloid cells in mouse models of breast cancer, and demonstrate that its inhibition within myeloid cells suppresses tumor growth by increasing intratumoral accumulation of effector CD8$^+$ T cells and immune-stimulatory myeloid subsets. Tumor-associated macrophages (TAMs) isolated from $Camkk2^{-/-}$ mice expressed higher levels of chemokines involved in the recruitment of effector T cells compared to WT. Similarly, in vitro generated $Camkk2^{-/-}$ macrophages recruit more T cells, and have a reduced capability to suppress T cell proliferation, compared to WT. Treatment with CaMKK2 inhibitors blocks tumor growth in a CD8$^+$ T cell-dependent manner, and facilitates a favorable reprogramming of the immune cell microenvironment. These data, credential CaMKK2 as a myeloid-selective checkpoint, the inhibition of which may have utility in the immunotherapy of breast cancer.

[1] Department of Medicine, Division of Hematological Malignancies and Cellular Therapy, Duke University School of Medicine, Durham, NC 27710, USA. [2] Department of Molecular Medicine and Medical Biotechnology, University of Naples Federico II, Naples 80131, Italy. [3] Molecular and Integrative Physiology, University of Illinois at Urbana-Champaign, Urbana, IL 61801, USA. [4] University of Illinois Cancer Center, Chicago, IL 60612, USA. [5] Cancer Center at Illinois, University of Illinois at Urbana-Champaign, Urbana, IL 61801, USA. [6] Department of Pharmacology and Cancer Biology, Duke University School of Medicine, Durham, NC 27710, USA. [7] Department of Biostatistics and Bioinformatics, Duke University, Durham, NC 27710, USA. [8] Department of Molecular and Cellular Biology, Baylor College of Medicine, Houston, TX 77030, USA. [9] Dan L. Duncan Cancer Center, Baylor College of Medicine, Houston, TX 77030, USA. [10] Department of Chemical Biology, GlaxoSmithKline, Research Triangle Park, NC 27709, USA. [11] CNR IFN Institute for Photonics and Nanotechnologies, Rome 00156, Italy. [12] Department of Population Sciences, City of Hope National Medical Center, Duarte, CA 91010, USA. [13] Department of Pathology, Duke University School of Medicine, Durham, NC 27710, USA. [14] Present address: Eli Lilly and Company, Indianapolis, IN 46285, USA. [15] Present address: UNC Eshelman School of Pharmacy, Chapel Hill, NC 27599, USA. Correspondence and requests for materials should be addressed to L.R. (email: luigi.racioppi@duke.edu) or to D.P.M. (email: Donald.McDonnell@duke.edu)

The recruitment of myeloid cells is an important process in the initial phases of tumor development. In response to cues from the tumor microenvironment, these cells can differentiate into tumor-associated macrophages (TAM), promoting angiogenesis, and supporting both primary tumor growth and distal metastasis[1,2]. TAMs are also responsible for the establishment of a robust immune-suppressive tumor ecosystem by inhibiting the differentiation and function of effector T cells, and stimulating the intratumoral accumulation of regulatory T cells (Tregs) and myeloid-derived suppressor cells (MDSCs)[3–5]. Not surprisingly, TAM density in primary tumors is strongly associated with poor outcomes in breast cancer[6–8]. These findings have driven the search for therapeutic targets, the manipulation of which will reprogram TAMs in a manner that promotes pro-immunogenic activities[9,10].

CaMKK2 is a $Ca^{2+}$/Calmodulin (CaM)-dependent serine–threonine protein kinase that couples calcium transients triggered by a variety of stimuli to processes involved in the regulation of cell proliferation, survival, and metabolism[11]. For the most part, these activities require CaMKK2-dependent phosphorylation of its primary substrates, CaMKI, CaMKIV, and/or AMPK[11,12]. The CaMKK–CaMKI pathway regulates the activity of multiple downstream targets, including ERK, RAC1, CREB, and HDAC5, which in turn impacts the cell cycle, cytoskeletal architecture, cell differentiation, as well as, facilitating hormone and cytokine production[13,14]. CREB is also a downstream target of CaMKIV, and activation of this signaling cascade promotes dendritic cell survival and increases cytokine secretion from T cells[15,16]. Both CaM and CaMKK2 can form a complex with AMPK subjecting the bound fraction of AMPK to regulation by calcium rather than AMP[13]. In addition to functioning as a key integrator/regulator of metabolic responses to energetic stress, AMPK has been shown to play a role(s) in the regulation of hematopoiesis, macrophage polarization, and T cell effector function.

The importance of CaMKK2 in the regulation of a number of inflammatory diseases such as obesity and insulin resistance and prostate and hepatocellular cancers has been established[11,12]. In prostate cancer cells, CaMKK2 expression is induced by androgens[17], and in hepatocytes by chemical carcinogens[18]. CaMKK2 is also expressed in gastric and ovarian cancers where pro-tumorigenic roles have been suggested[19,20]. To date, most studies of CaMKK2 in cancer have focused on its cancer cell intrinsic activities. However, this enzyme is also expressed within hematopoietic stem and progenitor cells, and in more mature myeloid cells, including monocytes and peritoneal macrophages[11,21–23]. Furthermore, genetic ablation of Camkk2 in mice revealed an important role for this enzyme in the development of myeloid cells and in regulating their ability to mount inflammatory responses to various stimuli[22,24]. These activities of CaMKK2 within myeloid cells suggested to us that it may also impact tumor biology in a cancer cell extrinsic manner. The goal of this study, therefore, was to investigate the extent to which CaMKK2 impacts immune cell repertoire and function in the microenvironment of mammary tumors. We find that deletion of CaMKK2 in myeloid cells, or its pharmacological inhibition, attenuates tumor growth in a $CD8^+$ T cell-dependent manner, facilitating a favorable reprogramming of the immune cell microenvironment. These data, credential CaMKK2 as a myeloid-selective checkpoint, the inhibition of which may have utility in the immunotherapy of breast cancer.

## Results

### CaMKK2 is expressed in tumor-associated stromal cells.
To probe the potential significance of CaMKK2 expression in human breast cancer, we analyzed CaMKK2 expression in two well-curated breast cancer tissue microarrays (Vienna and Roswell Park). CaMKK2 is found to be expressed in both cancer cells and within stromal cells (Fig. 1a; S1A). In the Vienna set, CaMKK2 expression inversely correlated with the less aggressive luminal A (LA) molecular type (OR = 0.2; $p$-value = 0.0241; Fisher's exact test), and a trend for higher CaMKK2 expression in triple-negative TN type was also observed (Table 1). The positive association between CaMKK2 expression and TN type was also found in the Roswell Park data set (OR = 5.0; $p$-value = 0.0129; Fisher's exact test). Analysis of the combined data confirmed the negative association of CaMKK2 expression with LA breast cancer (OR = 0.3; $p$-value = 0.017; Fisher's exact test) and elevated expression of CaMKK2 in TN tumors (OR 4.1; $p$-value = 0.0026; Fisher's exact test). Overexpression of CaMKK2 did not correlate with tumor grade (Supplementary Table 1), although a significant positive correlation between the expression of CaMKK2 in tumors and stromal cells was identified (Fig. 1b). Thus, CaMKK2 may contribute to tumor pathobiology through both cancer cell intrinsic and extrinsic actions.

The expression of CaMKK2 within immune cells that infiltrate mammary tumors was further assessed in a syngeneic model of breast cancer. Specifically, E0771 breast tumor cells were engrafted into the mammary fat pad of a [Tg(Camkk2-EGFP) C57BL/6 J] Camkk2-reporter mouse model[25,26], and the immune cell repertoire was analyzed by flow cytometry using a gating strategy (outlined in Supplementary Fig. 1B) that enables resolution of tumor-associated myeloid cell subsets[27]. While the expression of EGFP was minimal in lymphoid cells, it was readily detectable in monocytes, DCs and TAMs (Fig. 1c; Supplementary Fig. 1B). Thus, we hypothesized that CaMKK2 may impact the function of mammary tumor-associated myeloid subsets to influence tumor pathobiology.

### Attenuated growth of mammary tumors in Camkk2⁻/⁻ mice.
The impact of disrupting host CaMKK2 expression on mammary tumor growth was next evaluated. To this end, mammary tumors from MMTV-PyMT mice were grafted orthotopically into syngeneic Camkk2⁻/⁻ and wild-type (WT) mice. We found that MMTV-PyMT tumors were unable to grow within the mammary fat pad of Camkk2⁻/⁻ mice (Fig. 2a). These studies were repeated using the E0771 mammary tumor model, and a similar attenuation of tumor growth was observed when cells were engrafted in Camkk2 ablated hosts (Fig. 2b). Analysis of hematoxylin and eosin (H&E) and Masson's Trichrome stained tumors indicated that tumors propagated in Camkk2⁻/⁻ mice were more necrotic and fibrotic when compared with those grown in WT mice (Supplementary Fig. 2A). No significant difference in the number of $CD31^+$ endothelial cells was detected in tumors harvested from WT and Camkk2⁻/⁻ mice (Supplementary Fig. 2B).

As noted above, the Camkk2 promoter is highly active in myeloid cells, but not lymphoid cells within tumors. Thus, we reasoned that the decreased growth of mammary tumors observed in Camkk2⁻/⁻ mice might reflect an attenuation of immunosuppressive activities or an enhancement of immune-stimulatory functions of myeloid cells within tumors. An immunomorphometric analysis revealed that a significantly higher number of T cells ($CD3^+$) and macrophages ($F4/80^+$) accumulated in the non-necrotic areas of tumors from Camkk2⁻/⁻ mice (Fig. 2c, d). FACS analysis using a previously validated gating strategy[28] (outlined in Supplementary Fig. 2C) revealed fewer neutrophils and MHC II⁻ monocytes within tumors from Camkk2⁻/⁻ mice compared with their WT counterparts (Fig. 2e). A positive trend for accumulation of T and B cells in tumors of Camkk2⁻/⁻ mice were also observed. Lastly, the percentage of macrophages (including inflammatory monocytes) and DCs was significantly

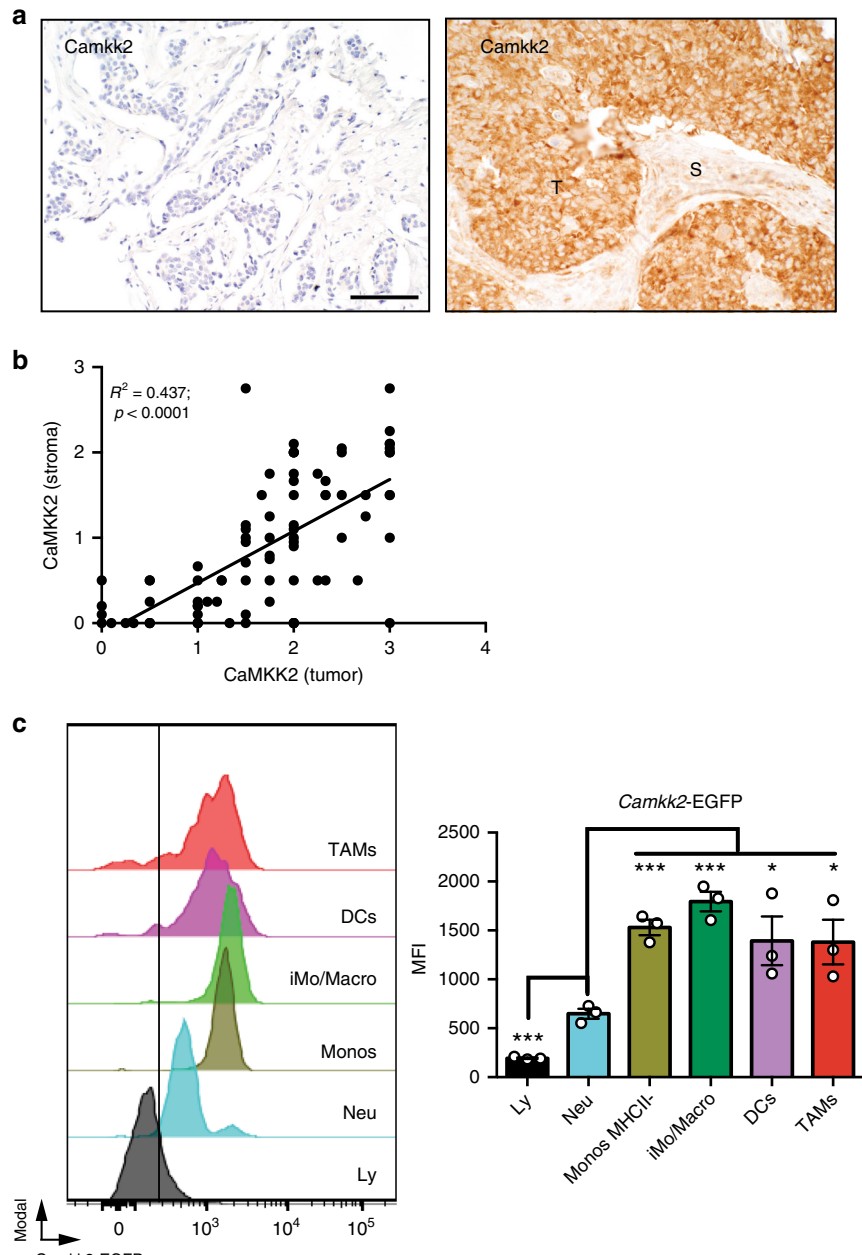

**Fig. 1** CaMKK2 is expressed in mammary tumor-associated stromal cells. **a** Expression of CaMKK2 in human breast cancer. Representative images (×400 magnification) of malignant mammary tissue samples stained with an anti-CaMKK2 antibody. T and S refer to tumor and stromal cells, respectively. Scale bar = 100 μm. **b** CaMKK2 staining intensity is correlated between cancer and stromal cells of the same human breast tumors. **c** The *Camkk2* promoter is active in myeloid cells associated with mammary tumors. E0771 cells (4 × 10$^5$ cells/mouse) were inoculated into the mammary fat pad of (Tg)-*Camkk2*-EGFP reporter mice. Subsequently, tumors were removed and digested and single-cell suspensions were stained for markers of myeloid cells. The gating strategy used to identify myeloid subsets and lymphoid cells is shown in Supplementary Fig. 1B. (Left) FACS profiles report the expression of EGFP reporter in tumor-associated immune cells. The vertical line refers to negative control (EGFP-negative splenocytes). (Right) Mean fluorescence intensity (MFI) of EGFP expression in immune cell subsets. Bar graph reports the mean ± SEM (standard error of the mean; N = 3 independent tumors in each group). *$p <$ 0.05, ****$p <$ 0.005, respectively. A *t* test was used to calculate *p*-values. Similar results have been observed in three independent experiments

higher in tumors propagated in *Camkk2*$^{-/-}$ mice when compared with WT mice (Fig. 2e).

Gene expression analysis was next undertaken to define how CaMKK2 knockdown impacted immune cell repertoire in tumors. We first confirmed the expression (qPCR and western immunoblot) of CaMKK2 in myeloid cells isolated from E0771 tumors (Supplementary Fig. 2D, E). Analysis of the whole-tumor lysates from *Camkk2*$^{-/-}$ mice showed a remarkable upregulation of genes expressed in the cytotoxic effector lymphocyte subsets, such

as Granzyme B (*Gzmb*) and Perforin-1 (*Prf1*) (Fig. 2f, top). Although expressed at very low levels, a significant upregulation of *Foxp3* and *Pdcd1* was also observed in tumors from *Camkk2*$^{-/-}$ mice. This may result from an increased accumulation of regulatory T cells in the more fibrotic/necrotic areas of tumors from *Camkk2*$^{-/-}$ mice. Notably, the expression of mRNAs encoding the CXCR3 receptor ligands (*Cxcl9*, *Cxcl10*, and *Cxcl11*), which have non-redundant roles in tumoricidal T cell trafficking[29], were found to be upregulated in tumors from *Camkk2*$^{-/-}$

**Table 1 CaMKK2 expression by molecular breast cancer subtypes**

| | Vienna[a] | | | | RPCI[b] | | | | Combined[c] | | | |
|---|---|---|---|---|---|---|---|---|---|---|---|---|
| CaMKK2 | N | LA | LB | TN | N | LA | LB | TN | N | LA | LB | TN |
| Low | 32 | 75% | 9% | 16% | 54 | 65% | 19% | 16% | 86 | 69% | 15% | 16% |
| High | 15 | 40% | 20% | 40% | 14 | 43% | 7% | 50% | 29 | 41% | 14% | 45% |
| OR[d] | | 0.2 | nd | ns | | ns | nd | 5.0 | | 0.3 | nd | 4.1 |
| p-value[e] | | 0.0241 | nd | ns | | ns | nd | 0.0129 | | 0.0107 | nd | 0.0026 |

N sample numbers, nd not determined, ns not significant

Immunohistochemical analysis of CaMKK2 expression in human breast cancer tissue microarrays, from two independent datasets ([8] and Roswell Park Cancer Institute, RPCI). CaMKK2 expression was determined to be low (<2) or high (>= 2), and correlated with molecular subtypes: triple negative (TN), luminal A (LA), and luminal B (LB). Fisher's exact test was used to determine p-values for the likelihood of association
[a]Vienna data set; 47 samples
[b]Roswell Park Cancer Institute data set; 68 samples
[c]Vienna and RPCI combined
[d,e]Odds ratio and associated p-value

mice compared with WT (Fig. 2f, bottom). Those genes found to be upregulated in tumors grown in *Camkk2*[−/−] mice (e.g., *Cxcl9* and *Gzmb*) were found to associate with a positive prognosis in human breast cancers (Supplementary Fig. 2F)[30].

Our IHC analysis revealed that more T cells accumulated in non-necrotic areas of tumors removed from *Camkk2*[−/−] mice compared with WT. FACS was used next to quantitatively assess activation and inhibitory markers expressed on CD4[+] and CD8[+] tumor-infiltrating lymphocytes (TILs) (gating strategy outlined in Supplementary Fig. 3A, B). Increased percentages of CD8[+] TILs expressing the activation markers GZMB and CD69 were detected in tumors of *Camkk2*[−/−] mice compared with WT (Fig. 3a, b). In contrast, tumors from WT mice demonstrated an increased accumulation of CD4[+] T cells expressing PD-1 and LAG-3 inhibitory receptors (Fig. 3b). Overall, these findings indicate that CaMKK2 knockdown promotes a more vigorous CD8[+] T cell-mediated immune response in tumors. Of significance, in breast cancer, the number of CD8[+] T cells in tumors positively correlates with better therapeutic response and prognosis[31,32].

Analysis of the tumor-associated myeloid cell repertoire revealed that CD11b[+] MHC II[+] cells accumulated preferentially in tumors from *Camkk2*[−/−] mice (Fig. 3c). This was also confirmed using tumors of similar sizes from *Camkk2*[−/−] and WT mice (Supplementary Fig. 2C and 3C). A large fraction of these cells also expressed the Ly6C marker, a phenotype indicative of inflammatory monocytes. A more detailed analysis confirmed an increase in MHC II[+] TAMs in tumors from *Camkk2*[−/−] mice (Supplementary Fig. 3C, top). Of note, MHC II[+] TAMs have previously been shown to exhibit M1-like features, and accumulation of this myeloid subset in tumors positively correlates with tumor regression[33,34].

Analysis of the tumor-infiltrating conventional dendritic cell compartment (TIDCs) identified MoDC as the largest fraction of cDCs infiltrating E0771 tumors, followed by cDC2 and cDC1 subsets (Supplementary Fig. 3C, bottom). Interestingly, increased numbers of MoDC were found in tumors from *Camkk2*[−/−] mice compared with WT (Supplementary Fig. 3C, bottom right). The role of TIDCs subsets in the tumor microenvironment is not fully understood[35,36], as both detrimental or even beneficial effects on CD8+ T cell activation have been reported for this myeloid subset[37,38].

**Tumor growth attenuation requires CD8[+] T cells and CaMKK2 inhibition in myeloid cells**. The extent to which the decreased tumor growth observed in *Camkk2*[−/−] mice could be attributed to the increased number of cytotoxic T cells within these tumors was next evaluated. *Camkk2*[−/−] mice were treated with an anti-CD8 antibody to deplete CD8[+] T cells or a control

IgG (Supplementary Fig. 4A, B). This intervention was accompanied by a dramatic increase in tumor growth in the *Camkk2*[−/−] background, but was without effect in WT mice (Fig. 3d; Supplementary Fig. 4C). These data confirmed that CD8[+] T cells are responsible, at least in part, for the impaired tumor growth observed in *Camkk2*[−/−] mice.

Since *Camkk2* promoter activity is restricted to the myeloid lineage in tumors (Fig. 1c), it seemed likely that CaMKK2 impacted tumor growth through its ability to regulate CD8[+] T cell function secondary to activities within myeloid cells. To test this possibility, we developed a LysMCre[+] *Camkk2*[fl/fl] mouse model and confirmed a reduced expression of CaMKK2 protein in macrophages of LysMCre[+] *Camkk2*[fl/fl] compared with LysMCre[+] *Camkk2*[wt/wt] littermates (Supplementary Fig. 4D). As observed in the whole-body knockout of *Camkk2*, the growth of E0771 tumors was significantly attenuated when propagated in LysMCre[+] *Camkk2*[fl/fl] mice compared with littermate controls, and this activity was associated with an increased accumulation of CD11b[+] MHC II[+] cells in tumors from LysMCre[+] *Camkk2*[fl/fl] mice (Fig. 3e; Supplementary Fig. 4E). These data were confirmed using LysMCre[−] *Camkk2*[fl/fl] mice and LysMCre[+] *Camkk2*[fl/fl] mice to rule out off-target effects of Cre expression on myeloid cell function (Supplementary Fig. 4F). Thus, we conclude that ablation of *Camkk2* within myeloid cells is sufficient to attenuate the growth of E0771 mammary tumors in immune-competent mice.

**CaMKK2 influences the expression of key genes in BMDM**. Cancer cell-secreted factors can influence myeloid cell differentiation resulting in an increase in the number/activity of TAMs and other immune-suppressive myeloid cell subsets[4,10]. Thus, we reasoned that genetic deletion of *Camkk2* might influence macrophage differentiation and/or activity in a manner that increases their immune-stimulatory phenotype. Analysis of the immune-regulatory cytokines produced by E0771 cells confirmed that, absent any provocative stimuli, they secreted high levels of VEGF, G-CSF, and CCL2 among others (Supplementary Fig. 5A, B). The impact of tumor-conditioned media (TCM) on myeloid cell function was next assessed using bone marrow cells isolated from WT and *Camkk2*[−/−] mice and differentiated in vitro in the presence of (a) regular media (RM) or (b) TCM (E0771) (Supplementary Fig. 5C).

The effects of TCM on the differentiation program of myeloid cells were evaluated using microarray analysis and confirmed by qPCR. Regardless of genotype, 439 differentially expressed genes (DEGs) were identified in bone marrow-derived myeloid cells (namely, BMDM—the purity assessed by FACS was >90% CD11b[+] F4/80[+], Ly6C[−], CD11c[dim/neg]) generated in the presence of TCM versus RM. Among these genes were several chemokine genes (*Ccl8*, *Ccl12*, *Cxcl10*, and *Cxcl11*) (Fig. 4a, b; Supplementary

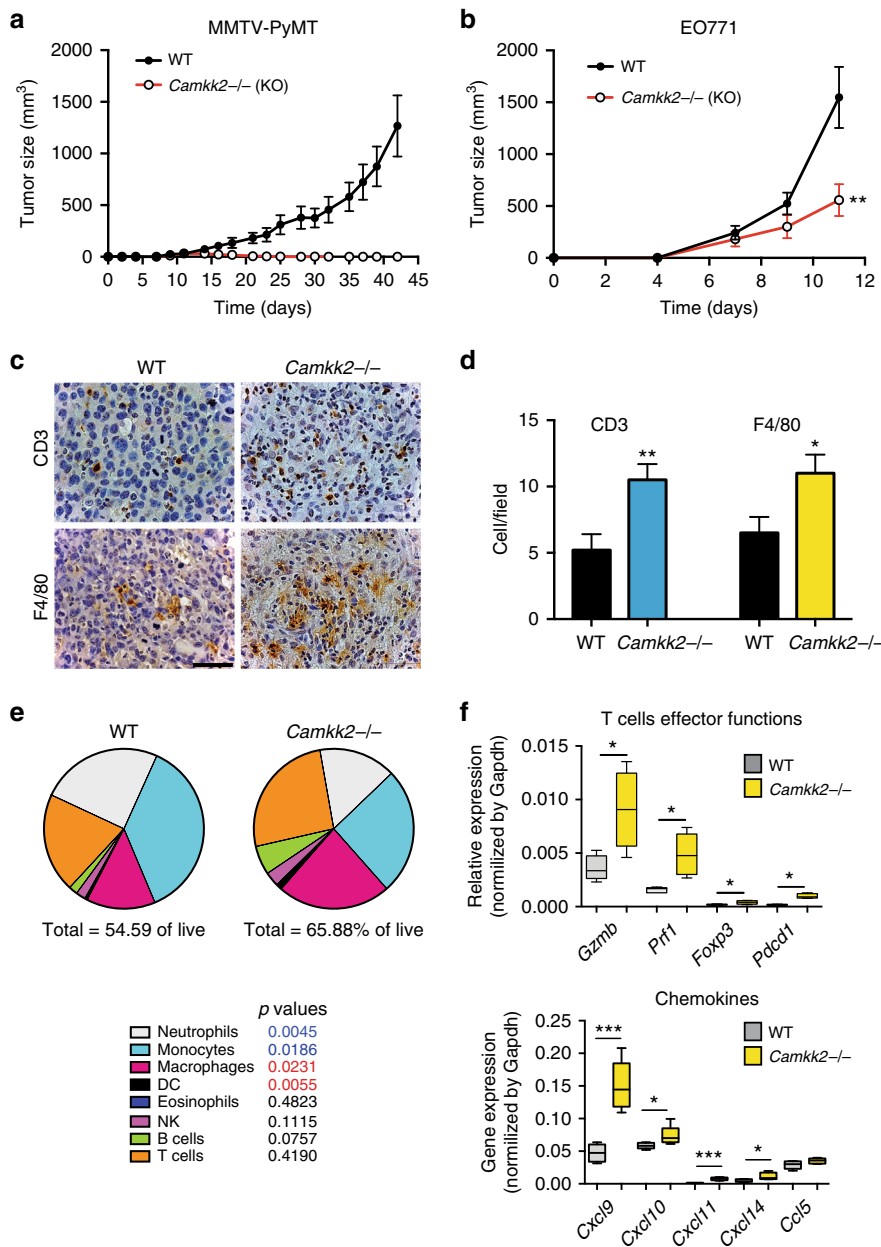

**Fig. 2** The growth of mammary tumors is attenuated in mice lacking CaMKK2. **a** Mammary tumors from MMTV-PyMT mice propagated in a C57BL/6 background were harvested, diced, and orthotopically grafted into syngeneic C57BL/6 mice that were WT or knockout for Camkk2 (WT and Camkk2$^{-/-}$, respectively; mean ± SEM; N = 10 in each group). **b** Murine E0771 cells (4 × 10$^5$ cells/mouse) were orthotopically grafted in WT and Camkk2$^{-/-}$ mice, and subsequent tumor volume measured as indicated (N = 5 in each group; this experiment was replicated with a comparable number of mice at least three times). A two-way ANOVA test was used to calculate p-values. **c** Representative images of E0771 tumors from WT and Camkk2$^{-/-}$ mice. Sections were stained with anti-CD3 or anti-F4/80 antibodies (scale bar = 50 μm). **d** Quantitation of CD3$^+$ and F4/80$^+$ cells in high-power optic (×400 magnification) fields in stained sections (six fields for each section; N = 3 individual tumors in each group). Bar graph reports the mean ± SEM. N = 18 data points/ genotype. *$p < 0.05$, **$p < 0.01$, respectively. A t test was used to calculate p-values. **e** E0771 tumors of comparable size (500–700 mm$^3$) were removed from WT and Camkk2$^{-/-}$ mice, digested with collagenase and DNase I to obtain a single-cell suspension. Immune cells were identified by flow cytometry, according to the gating strategy reported in Supplementary Fig. 2C. Pie charts show the mean percentages of immune cell subsets detected in tumors removed from WT and Camkk2$^{-/-}$. Values reported under each pie chart refer to the percentages of tumor-infiltrating CD45+ cells. N = 8 and 4 tumors from WT and Camkk2$^{-/-}$ mice were analyzed. A t test was used to calculate statistical significance. p-values color code: blue and red values indicate a statistically significant increase or decrease of the indicated cell types in tumors from WT compared with KO host, respectively. **f** Gene expression analysis in E0771 tumors removed from WT and Camkk2$^{-/-}$ mice (N = 5 and 5, respectively). A t test was used to calculate p-values. Bar graph reports the mean ± SEM. *$p < 0.05$, ***$p < 0.005$

Table 2). Furthermore, 455 genes were selectively regulated in Camkk2$^{-/-}$ TCM-BMDM compared with WT TCM-BMDM (Supplementary Fig. 6A), and these DEGs were associated with 24 canonical pathways (Supplementary Data 1). Most of the genes associated with steroid biosynthesis (KEGG 00100; p-value 4.236e-4; Fisher's method) were downregulated in Camkk2$^{-/-}$ TCM-BMDM compared with WT (Supplementary Fig. 6B). In contrast, several DEGs associated with cytokine–cytokine receptor pathways

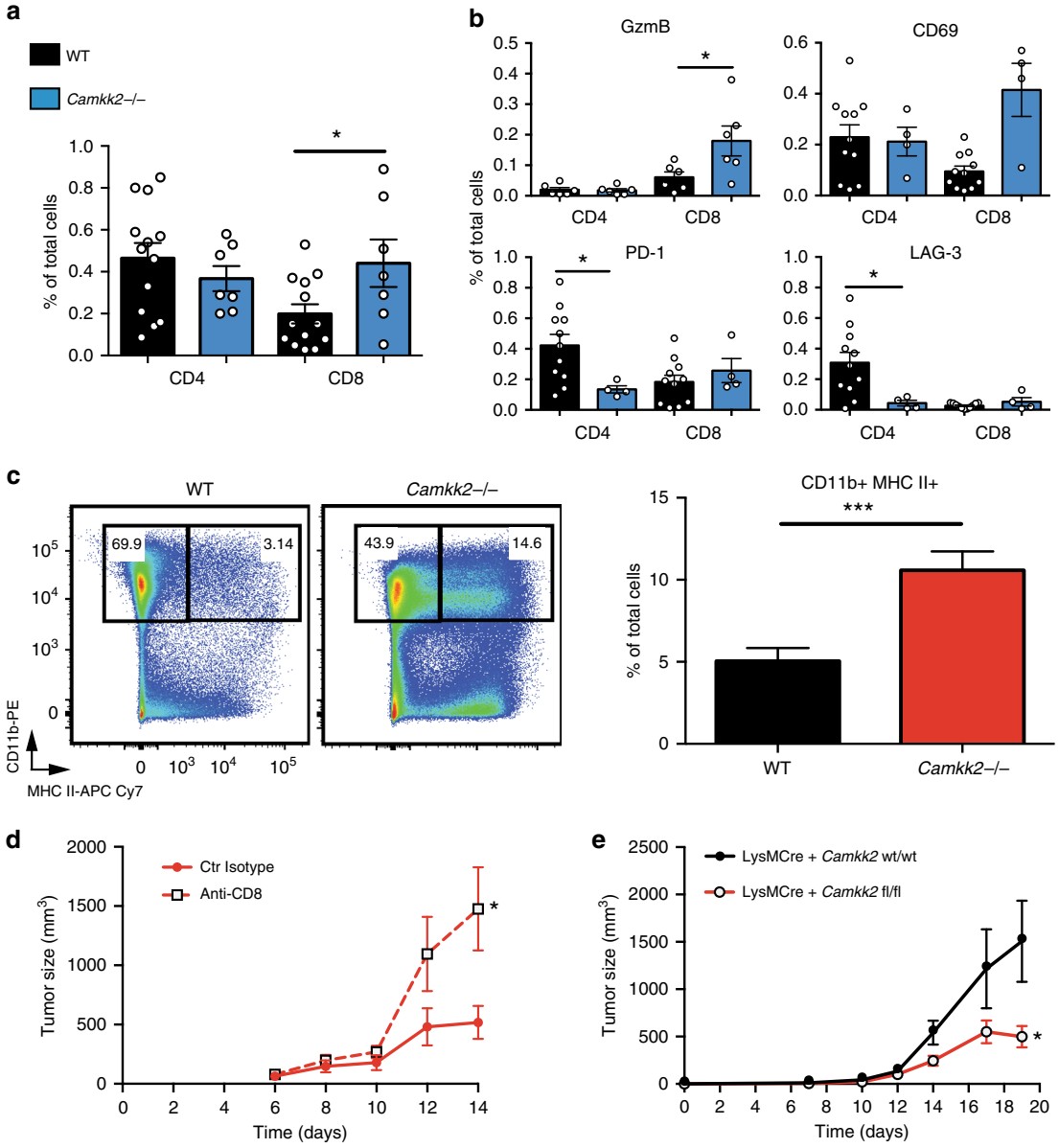

**Fig. 3** Attenuated tumor growth in *Camkk2* KO host is mediated by CD8+ T cells. Murine E0771 ($4 \times 10^5$) cells were orthotopically grafted in WT and *Camkk2*−/− mice, and subsequently tumors of comparable sizes (500–700 mm³) were harvested and digested with both collagenase and DNase I. Single-cell suspensions were then stained for lymphoid and myeloid markers, and cell subsets were identified according with gating strategy reported in Supplementary Fig. 3A–C. **a** Increased accumulation of CD8+ T cells in E0771 cell-derived tumors propagated in *Camkk2*−/− mice compared with WT. Bar graph reports the mean ± SEM from two combined independent experiments. **b** Percentages of tumor-infiltrating lymphocyte subsets in E0771 tumors growing in *Camkk2*−/− mice compared with WT. (upper left panel) Intracellular expression of Granzyme B+ (GzmB+). CD69+, PD-1+, and LAG-3+ expression was detected on an independent set of eight and four tumors removed from WT and *Camkk2*−/− mice, respectively. Bar graph reports the mean ± SEM. A *t* test was used to calculate *p*-values. **c** Representative FACS dot plots of tumor single-cell suspensions stained with myeloid markers (left). Bar graph reports the mean ± SEM of CD11b+ MHC II+ subset from combined independent experiments. A *t* test was used to calculate *p*-values. *N* = 13 and 15 tumors removed from WT and *Camkk2*−/− mice, respectively. **d** Mammary tumor growth in *Camkk2*−/− mice depleted of CD8+ T cells. *Camkk2*−/− mice were treated with anti-CD8 antibody or control isotype every 3 days starting a week before tumor cell inoculation. At day 0, E0771 cells were orthotopically grafted, and mice were treated with antibodies every 4 days. Graph depicts tumor size (mean ± SEM; *N* = 6 in each group). **e** Mammary tumors display retarded growth in mice devoid of *Camkk2* in myeloid cells. E0771 cells were orthotopically grafted into LysMCre+ *Camkk2*wt/wt and LysMCre+ *Camkk2*fl/fl mice. Tumor volume was measured (mean ± SEM; *N* = 4 for each group). This experiment was replicated with similar results. A two-way ANOVA test was used to calculate *p*-values. **p* < 0.05, ***p* < 0.005

(KEGG 04060; *p*-value 4.236e-4; Fisher's method; Supplementary Fig. 6B, Supplementary Data 2) were upregulated in *Camkk2*−/− TCM-BMDM compared with WT. Using gene set enrichment analysis (GSEA)[39], a positive enrichment for interferon response genes and negative enrichment of genes involved in cholesterol biosynthesis were noted in *Camkk2*−/− TCM-BMDM compared with WT TCM-BMDM (Supplementary Fig. 6C, D).

We reasoned that ablation of *Camkk2* would prompt myeloid progenitors exposed to TCM to develop toward a more immunogenic phenotype compared with those derived from

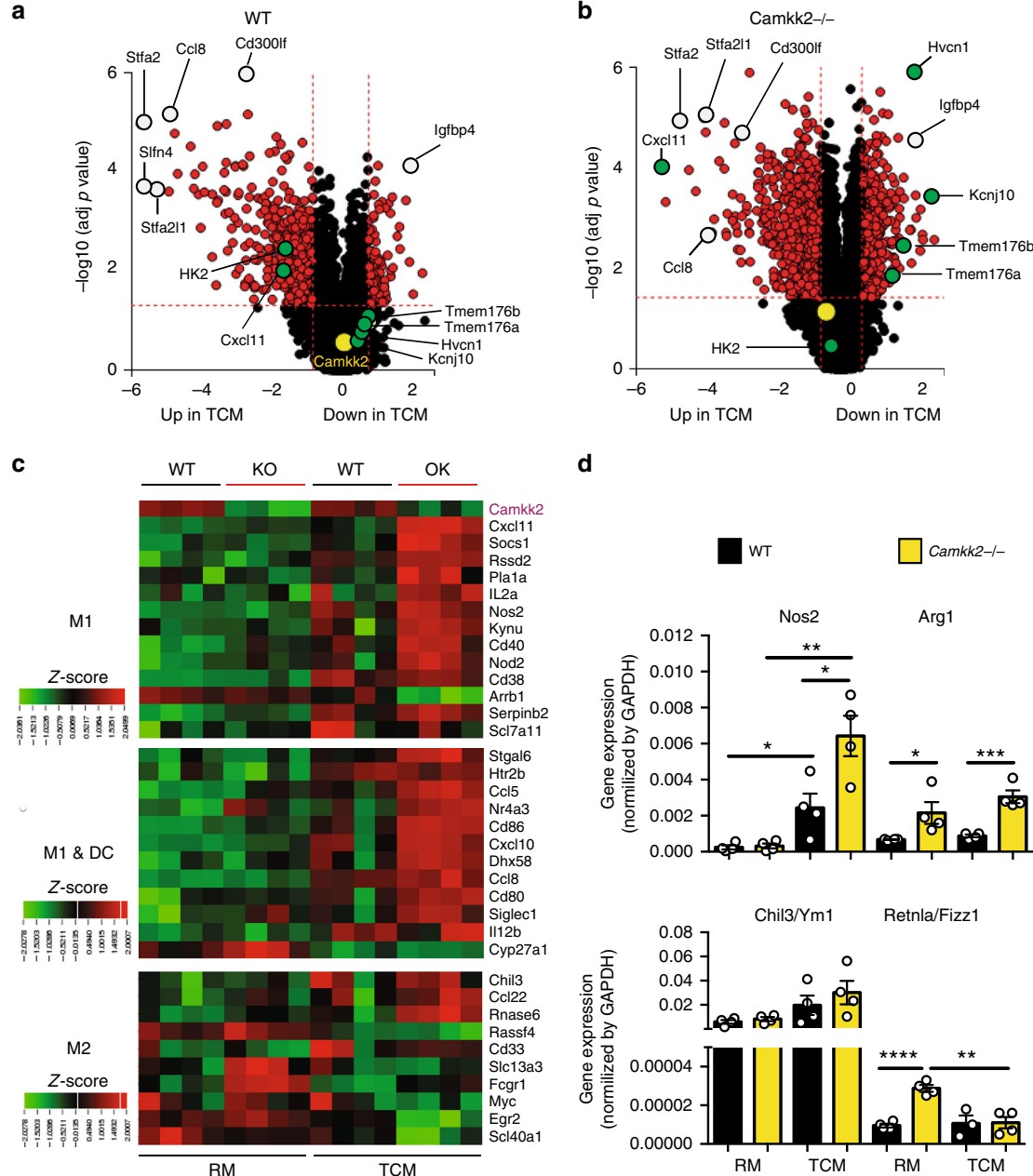

**Fig. 4** CaMKK2 regulates gene transcription in BMDM. Bone marrow-derived macrophages (BMDM) were generated from WT and $Camkk2^{-/-}$ mice in the presence or absence of E0771-conditioned medium (RM and TCM, respectively). Subsequently, BMDM was harvested and analyzed for gene expression. $N = 4$ biological replicates/group. **a**, **b** Volcano plots of differentially expressed genes (DEGs) in WT (left) and $Camkk2^{-/-}$ (right) generated in the presence of RM or TCM. Open circles indicate genes showing comparable expression levels in WT and $Camkk2^{-/-}$ BMDM. DEGs between WT and $Camkk2^{-/-}$ BMDM are shown as green filled circles. Yellow open circles refer to the $Camkk2$ gene. **c** Heatmaps of DEGs affiliated with M1, M1 and dendritic cells (M1&DC), or M2 signatures. The color key for the heatmap indicates (row-wise) scaled RPKM values (z-score). **d** Real-time quantitative PCR (qPCR) analysis of genes associated with M1 ($Nos2$) and M2 ($Arg1$, $Chil3/Ym1$ and $Retnla/Fizz1$) gene signatures (four biological replicates were analyzed for each genotype). A $t$ test was used to calculate $p$-values. The data are graphed as the mean ± SEM. Asterisks refer to $*p < 0.05$, $**p < 0.01$, $***p < 0.005$, and $****p < 0.001$. This experiment was replicated at least three times with similar results

WT mice. We therefore compared the expression of genes, previously shown by others to be associated with M1, shared by M1 and DCs (M1&DC), or M2 phenotypes[40], in WT and $Camkk2^{-/-}$ BMDM generated in the presence of RM and TCM. The most remarkable finding in this analysis was that the majority of M1 and M1&DC signature genes were upregulated in $Camkk2^{-/-}$ versus WT BMDM in the presence of TCM (Fig. 4c). The expression pattern of genes associated with the M2-phenotype was more complex with some genes also being

significantly upregulated in $Camkk2^{-/-}$ BMDM cultured in TCM (Fig. 4c). Finally, we used qPCR to evaluate the expression of $Nos2$, $Arg1$, $Chil3/Ym1$, and $Retnla/Fizz1$, genes used as surrogates of M1 or M2 polarization (Fig. 4d). A dramatic upregulation of $Nos2$ expression in $Camkk2^{-/-}$ compared with WT BMDM (Fig. 4d) was noted. Increased expression of $Arg1$ and $Retnla/Fizz1$ was also observed in $Camkk2^{-/-}$ BMDM compared with WT. Although the enhanced expression of $Retnla/Fizz1$ and $Arg1$ can be associated with an immunosuppressive

phenotype, when considered in total, these findings indicate that deletion of CaMKK2 interfered with the expression of the largely immunosuppressive transcriptional program induced by tumor-derived factors (TCM) in myeloid cells. Further CaMMK2 inhibition enhanced the transcription of genes associated with a more immunogenic phenotype, including interferon response genes and chemokines involved in intratumoral T cell trafficking.

**CaMKK2 links tumor factor signaling to AMPK activation.** Our data suggest that CaMKK2 is required to couple the proximal signaling events induced by tumor-derived factors present in the TCM with downstream molecular effectors. CaMKIV was eliminated from this analysis[41], as we were unable to detect CaMKIV/phospho-CaMKIV expression within BMDM under the conditions used for these in vitro assays. Comparable levels of phospho-CaMKI were detected in RM-BMDM and TCM-BMDM, from both WT and $Camkk2^{-/-}$ mice, indicating that tumor-derived factors are not increasing all aspects of CaMKK2 signaling in BMDM (Fig. 5a). AMPKα can be phosphorylated by both CaMKK2 and LKB1; however, we found it to be phosphorylated to a lesser degree in $Camkk2^{-/-}$ BMDM compared with WT BMDM when cultured in regular media (RM) (Fig. 5b, c). More importantly, increased AMPKα phosphorylation was observed in WT BMDM when cultured in TCM compared with RM, and this enhanced phosphorylation was attenuated considerably by CaMKK2 ablation (Fig. 5b, c). In aggregate, these data indicate that CaMKK2 is required for TCM-induced activation of AMPK cascade. Interestingly, in the absence of CaMKK2, the exposure to TCM displayed an inhibitory effect on the phosphorylation of AMPK. This finding is suggestive of a wider, yet undefined, function of CaMKK2 in processes that link tumor factor-induced signaling to the activation of AMPK. These data are interesting in light of those from others, indicating that AMPKα modulates macrophage polarization toward an M2-like anti-inflammatory phenotype[42–46]. Previously, it was determined that the PGC1/ERRα complex, a downstream target of AMPKα in macrophages, is a key regulator of innate immunity[47]. However, we ruled out the involvement of this complex in CaMKK2 biology as myeloid cell-specific knockdown of ERRα failed to impact the growth of mammary tumors in the MMTV model (Supplementary Fig. 7A–D).

**CaMKK2 controls the immune-stimulatory functions of TCM-BMDM.** The multi-ligand endocytic mannose receptor (CD206/MRC1) is an established marker of M2-like macrophages, whereas higher levels of MHC II and/or co-stimulatory molecules (e.g., CD80, CD86, and CD40) serve as markers of more immunogenic M1-like macrophages[4,33]. Thus, we analyzed the expression of these markers in WT and $Camkk2^{-/-}$ BMDM generated in the presence or absence of TCM (Fig. 6a, c; Supplementary Fig. 8A). Regardless of genotype, an increase in the percentage of CD206+ MHC II− (M2-like macrophages) and a decrease in the percentage of CD206− MHC II+ (M1-like macrophages) were detected in BMDM generated in the presence of TCM (Fig. 6a). These changes were attenuated in $Camkk2^{-/-}$ BMDM, such that in the presence of TCM they expressed higher levels of MHC II molecules and have fewer M2-like and more M1-like cells compared with WT (Fig. 6a, b). $Camkk2^{-/-}$ BMDM also expressed significantly higher levels of MHC II and CD40 compared with WT, although its absolute expression level was reduced in response to TCM (Fig. 6b; Supplementary Fig. 8A). BMDM generated in the presence of TCM also expressed higher levels of CD80 and MHC I molecules, although these TCM-induced changes were attenuated in $Camkk2^{-/-}$ BMDM. Finally, CD86 was expressed at comparable levels in WT and $Camkk2^{-/-}$

BMDM in the presence of RM, but higher levels of this co-stimulatory molecule were detected in $Camkk2^{-/-}$ BMDM generated in TCM when compared with WT (Supplementary Fig. 8A). To evaluate the effects of TCM on already differentiated BMDM, cells were first cultured in the presence of RM for 4 days, and then exposed for an additional 48 h to TCM. These experiments confirmed the ability of TCM to increase the percentage of M2-like cells in WT BMDM, but importantly this response was attenuated in $Camkk2^{-/-}$ BMDM (Supplementary Fig. 8B). One of the most important findings from these experiments was that the expression of RNAs encoding chemokines involved in the recruitment of immune cells (Cxcl9, Cxcl10, and Cxcl14) in tumors are increased in $Camkk2^{-/-}$ TCM-BMDM compared with WT TCM-BMDM (Fig. 6c; Supplementary Fig. 8C).

The significance of data generated in BMDM was subsequently validated in TAM isolated from E0771 tumors propagated in WT and $Camkk2^{-/-}$ mice. Similar to the results obtained in BMDM, deletion of CaMKK2 in the host was associated with a decreased percentage of CD206+ MHC II− TAM, and increased accumulation of CD206− MHC II+ TAMs (Fig. 6d). Of note, increased levels of Cxcl9 and Cxcl14 RNAs were also found in TAM isolated from tumors of $Camkk2^{-/-}$ mice, confirming the results of studies performed in vitro (Fig. 6e; Supplementary Fig. 8D).

The ability of WT and $Camkk2^{-/-}$ BMDM to stimulate and recruit T cells was next assessed. Syngeneic T cells were isolated from the spleens of WT mice, stained with the fluorescent dye 5-(and 6-)carboxyfluorescein diacetate succinimidyl ester (CFSE), and subsequently co-cultured with BMDM, in the presence of an anti-CD3 antibody. After 72 h, cell supernates were collected and assayed for cytokine expression, and T cell proliferation was assessed by FACS (Fig. 7a, b; Supplementary Fig. 8E). IL-2 production was significantly increased in T cells co-cultured with $Camkk2^{-/-}$ BMDM generated in TCM, but not RM (Supplementary Fig. 8E, left panel). Not unexpectedly, IFNγ expression was dramatically suppressed when T cells were cultured with WT BMDM generated in the presence of TCM (Supplementary Fig. 8E, right panel). However, this suppressive effect of TCM was mitigated upon CaMKK2 knockdown (Supplementary Fig. 8E, right panel). The functional importance of these differences was highlighted in follow-up T cell proliferation studies, which demonstrated that BMDM derived from WT or $Camkk2^{-/-}$ mice in RM were equally effective at stimulating T cell proliferation. However, T cell proliferation was reduced by ~50% when evaluated in the presence of WT TCM-BMDM, whereas deletion of $Camkk2$ in BMDM reduced this effect (Fig. 7a, b). We next explored the mechanisms by which CaMKK2 regulates the functional interactions between BMDM and T cells. Specifically, the role(s) of physical cell contact vs BMDM-released soluble factors on the proliferation of purified CD4+ and CD8+ T cells was assessed. To this end, purified CFSE-labeled CD4+ and CD8+ T cells were cultured with WT or $Camkk2^{-/-}$ TCM-BMDM, in the presence of an anti-CD3 antibody. In this direct cell contact assay, $Camkk2^{-/-}$ TCM-BMDM showed an increased ability to stimulate both purified T cell subsets, compared with WT TCM-BMDM. CFSE-labeled purified CD4+ and CD8+ T cells were then cultured with anti-CD3/anti-CD28 coated beads in the presence of supernates collected from TCM-BMDM cultures (Supplementary Fig. 8G). Interestingly, WT TCM-BMDM supernates inhibited the proliferation of CD8+ and CD4+ T cells while no inhibitory effect was observed using $Camkk2^{-/-}$ TCM-BMDM supernates (Supplementary Fig. 8G). Taken together these data suggest that the positive effect of $Camkk2^{-/-}$ TCM-BMDM on T cell proliferation likely requires physical contact, and this activity is reinforced by a reduced production of immune-suppressive soluble factors.

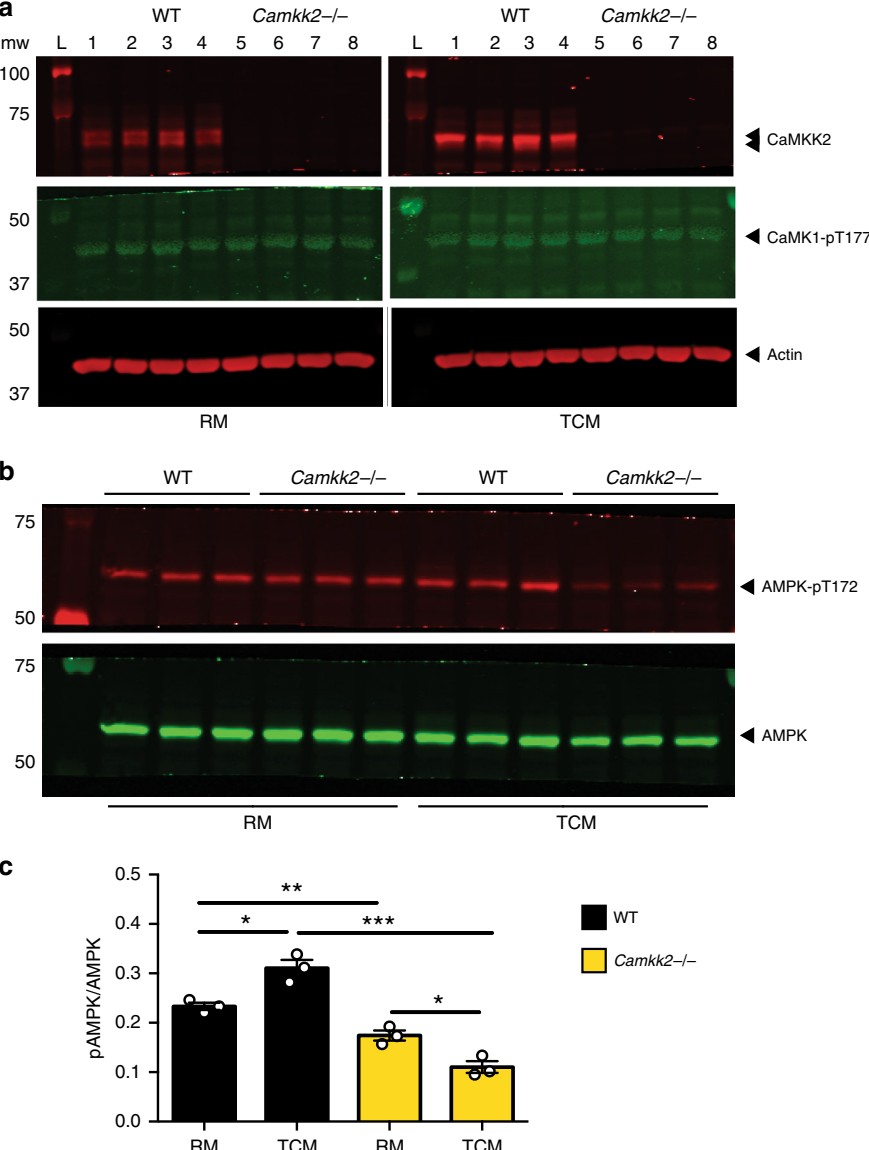

**Fig. 5** CaMKK2 links tumor factor signaling to AMPK activation. WT and Camkk2$^{-/-}$ BMDM were generated in regular differentiation medium in the presence or absence of E0771-conditioned medium (RM and TCM, respectively). **a** CaMKK2 and phospho-CaMK1 immunoblots. **b** Expression of phosphorylated (T-172; p-AMPK) and the total AMPKα was assessed by immunoblot. **c** Quantification of p-AMPK/AMPK ratio ($N = 3$ biological replicates; mean ± SEM). Each lane refers to protein lysate extracted from a single biological replicate. A $t$ test was used to calculate $p$-values. Asterisks refer to *$p < 0.05$, **$p < 0.01$, and ***$p < 0.005$, respectively

Gene expression analysis in whole tumors, isolated TAM, and BMDM identified the CXCR3-ligands (Cxcl9, Cxcl10, and Cxcl11) as relevant targets of CaMKK2 in myeloid cells. Specifically, deletion of Camkk2 resulted in increased expression of the mRNAs encoding Cxcl9 and Cxcl10 in TAMs and BMDM; a finding that could explain the increased accumulation of CD8[+] T cells in tumors of Camkk2$^{-/-}$ mice. To test this possibility, we developed a transwell assay to evaluate the ability of BMDM to attract in vitro-activated T cells. It was determined that more T cells migrated toward supernates collected from Camkk2$^{-/-}$ TCM-BMDM than those from WT mice (Fig. 7c, left). This conclusion was supported by additional data from a second study using antigen-specific CD8[+]T cells (OTI T) (Fig. 7c, right).

To further investigate the ability of WT and Camkk2$^{-/-}$ myeloid cells to recruit effector T cells in the tumor micro-environment, we used a validated microfluidic platform to recreate interconnected 3D spaces[48]. This device combines the

design of a hydrogel-based microchannel platform with advanced microscopy, allowing single-cell, real-time monitoring, and analysis of antigen-specific CD8[+] T cell migration. It further allows the monitoring of interactions that occur between effector T cells and cellular components of the tumor microenvironment. To build an on-chip simulation of the tumor microenvironment, BMDMs were co-cultured in a 3D hydrogel-based compartment with E0771 cells in the presence of OVA$_{257-264}$ peptide. The ability of WT and Camkk2$^{-/-}$ myeloid cells to recruit OVA-specific EGFP-CD8[+] T cells (EGFP-OTI-T) was tested under competitive conditions. To this end, the two tumor chambers of the microfluidic platforms included WT or Camkk2$^{-/-}$ BMDM (right and left chamber, respectively; Supplementary Fig. 9A, B). Upon hydrogel polymerization, EGFP-OTI-Ts were loaded and uniformly distributed into the immune chamber alongside with OVA$_{257-264}$-preloaded DsRed bone marrow-generated dendritic cells (DsRed-BMDC). The competitive ability of tumor chambers

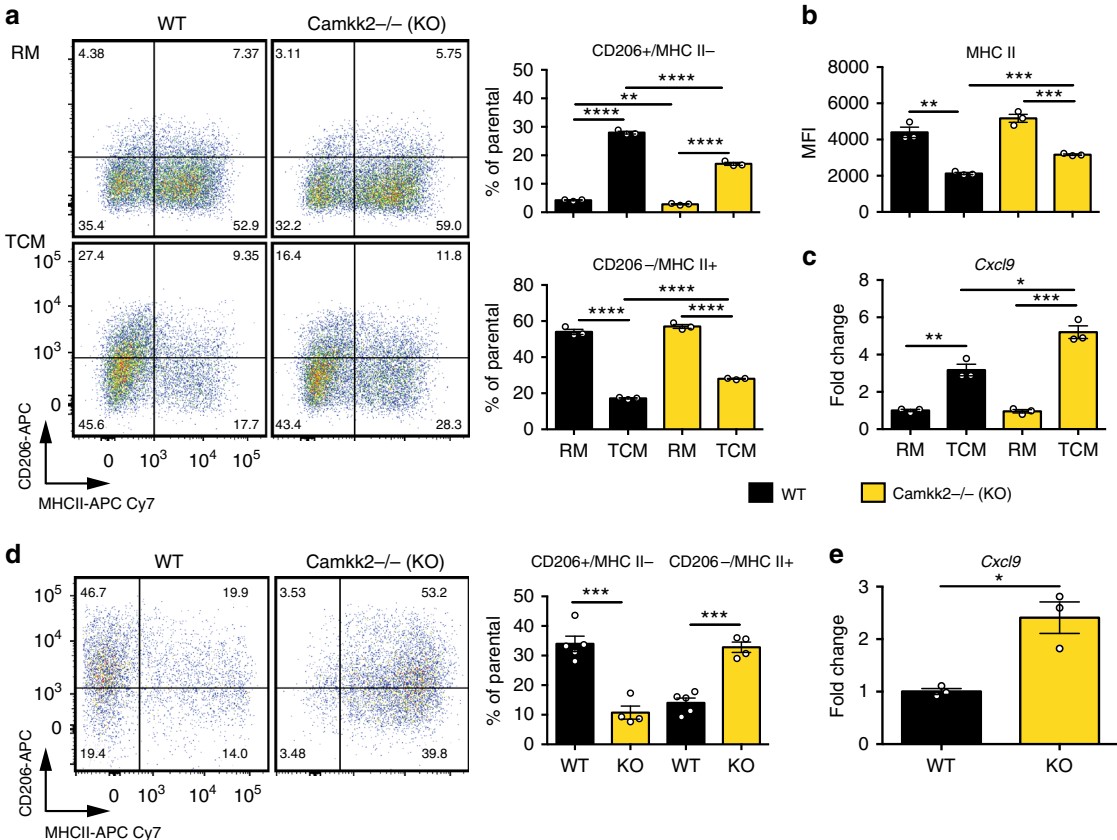

**Fig. 6** Phenotype of WT and *Camkk2*⁻/⁻ BMDM generated in the presence or absence of tumor-conditioned medium. WT and *Camkk2*⁻/⁻ BMDM were generated in the presence of RM or TCM. **a** Left: CD206 and MHC II expression on CD11b⁺ F4/80⁺ gated BMDM. Right: percentages of CD206⁺/MHC II⁻ and CD206⁻/MHC II⁺ subsets (mean ± SEM; $N = 3$ biological replicates). **b** Expression of MHC II on CD11b + F4/80 + gated BMDM (mean of MFI ± SEM; $N = 3$ biological replicates). **c** *Cxcl9* expression (mean ± SEM; $N = 3$ biological replicates). These experiments were replicated at least three times. **d** Tumors of comparable size (500–700 mm³) were removed from WT and *Camkk2*⁻/⁻ mice, and myeloid cells were then identified and sorted by flow cytometry. Bar graph reports mean ± SEM of CD206⁺/MHC II⁻ and CD206⁻/MHC II⁺ TAM percentages gated on "Mac" subset in the Supplementary Fig. 2 ($N = 5$ and 4 tumors removed from WT and *Camkk2*⁻/⁻ mice, respectively). **e** *Cxcl9* expression in TAM sorted from E0771 tumors removed from WT and *Camkk2*⁻/⁻ mice (mean ± SEM; $N = 3$ and 3 tumors for each genotype)

containing WT and *Camkk2*⁻/⁻ BMDM was evaluated over a period of 72 h by wide-field imaging analysis of the entire microfluidic device, and the total number of OTI T cells within the microchannels and tumor chambers was quantified (Fig. 7d; Supplementary Fig. 9C). OTI T cells initially interacted with antigen-loaded BMDC in the central chamber (1–6 h) and then migrated through micro-channels and 3D tumor compartments at later time points (18–44 h). Under these conditions, a remarkably higher number of OTI T cells migrated through the micro-channels and infiltrated the 3D tumor chamber containing *Camkk2*⁻/⁻ BMDM, compared with those moving toward the tumor chamber housing WT BMDM (Fig. 7d). Taken together, these studies define an important role for CaMKK2 in the regulation of cellular processes involved in the trafficking of T cells into tumors.

**CaMKK2 inhibitors attenuate mammary tumor growth.** STO-609 is a selective CaMKK2 inhibitor that has previously been validated for use in vivo[17,18,49]. Thus, we used this compound to evaluate the impact of CaMKK2 inhibition on the growth of E0771, 4T1, and Met1 cell-derived tumors propagated in syngeneic hosts, as well as in the MMTV- PyMT spontaneous model of mammary carcinoma (Fig. 8a; Supplementary Fig. 10A–C). Given the positive impact of CaMKK2 inhibition by STO-609 on tumor growth, we proceeded to generate novel, chemically

distinct, CaMKK2 inhibitors via a discovery campaign that led to the identification of GSK1901320 (denoted GSKi; Supplementary Fig. 10D). This compound is chemically distinct from STO-609, but shows similar antagonist activity to this drug in biochemical assays (Supplementary Fig. 10D). Importantly, GSKi inhibited the growth of E0771-derived tumors in syngeneic hosts to the same degree as STO-609 (Supplementary Fig. 10E). Thus, two chemically distinct, selective CaMKK2 antagonists demonstrated similar inhibitory effects in mouse models of breast cancer. These results justify further exploration/optimization of highly selective CaMKK2 inhibitors as a means to increase tumor immunity.

The impact of CaMKK2 inhibition on immune cell repertoire/function within tumors was next evaluated. Reflecting what was seen in genetic CaMKK2 knockout models, treatment with STO-609 promoted the accumulation of CD8⁺ and NK1.1⁺ T cells, CD11b⁺ MHC II⁺ cells, and MHC II⁺ TAMs within E0771 tumors (Fig. 8b, c; Supplementary Fig. 11A–C). STO-609 was without effect in CD8⁺ T cell depleted *Camkk2*⁻/⁻ mice, suggesting that its actions in primary tumor growth are likely limited to immune cells (Fig. 8d).

## Discussion
Our findings highlight the role(s) of CaMKK2 as a myeloid-expressed regulator that integrates signals controlling activation and differentiation of macrophages in the tumor microenvironment.

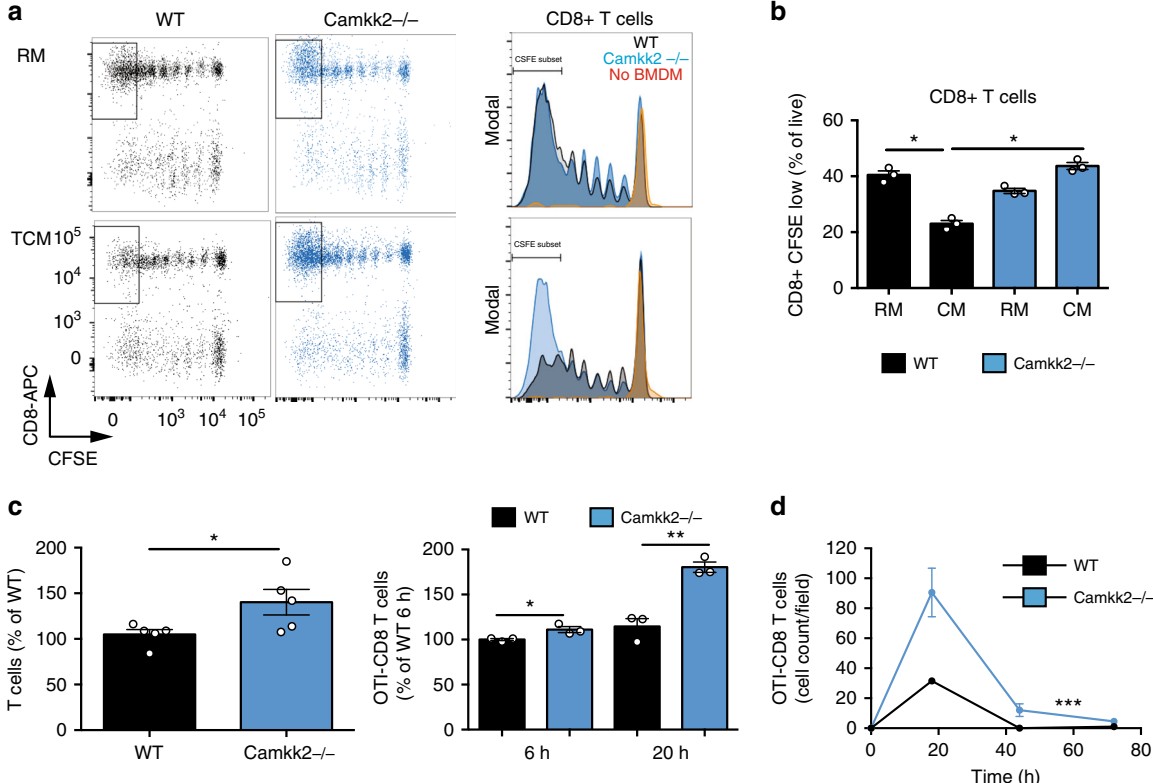

**Fig. 7** CaMKK2 mediates the immune-stimulatory ability of BMDM generated in tumor-conditioned medium. **a** CFSE-labeled T cells isolated from wild-type mice were cultured with anti-CD3 antibody, in the presence or absence of WT or Camkk2$^{-/-}$ BMDM generated in regular medium or the tumor-conditioned medium (RM and TCM, respectively). Dot plots and histograms of T cells recovered after co-culture for 72 h (Left and right, respectively). T cells were identified according to the gating strategy reported in the Supplementary Fig. 3A. **b** Percentage of CFSE-low CD8$^+$ T cells (gate is displayed in the panel A dot plots). $N = 3$ biological replicates for each genotype. $t$ test was used to calculate $p$-values. The experiment was replicated with similar results. **c** BMDM were generated from WT and Camkk2$^{-/-}$ mice (WT and KO, respectively) in the presence of TCM. After 5 days of culture, BMDM supernates were collected and tested in a chemotaxis transwell assay for the ability to recruit T cells isolated from lymph nodes of WT mice or CD8+ OTI T cells (left and right panels, respectively). Bar graphs report mean ± SEM of T cells migrating toward BMDM supernates after 12 h and percentages of CD8 + OTI T cells migrating toward BMDM supernates at 6 and 12 h (mean ±SEM; $N = 3$ replicates for each genotype). $t$ test was used to calculate $p$-values. These experiments were replicated with similar results. **d** A 3D microfluidic tumor on-a-chip model was used to test the effects of BMDM in E0771 tumor microenvironment (schematic of 3D on-chip model is shown in Supplementary Fig. 9). Graph reports kinetics of OTI CD8$^+$ T cells migrating toward 3D E0771-microenvironment infiltrated by WT or Camkk2$^{-/-}$ BMDM (mean± SEM; N = 3 low magnification fields). Two-way ANOVA was used to calculate $p$-values. Asterisks refer to *$p < 0.05$, **$p < 0.01$, ***$p < 0.005$, and ****$p < 0.001$, respectively

Importantly, genetic ablation of CaMKK2, or its pharmacological inhibition led to accumulation of less immune-suppressive myeloid cells and more activated CD8$^+$ T cells in the tumor microenvironment, which results in tumor growth inhibition. Mechanistically, our studies reveal that CaMKK2 is required to link tumor-derived factor signaling with the activation of AMPK, which likely functions to impair the immune-stimulatory functions of myeloid cells within the TME[44–46].

Previously, we reported that the *Camkk2* promoter is active in normal circulating monocytes, and that this enzyme is expressed in peritoneal macrophages[24]. Here, we demonstrate that CaMKK2 is also expressed in tumor-associated myeloid cells, as well as in human breast cancer cells. Importantly, our findings indicate that *Camkk2* promoter activity is differentially regulated throughout myeloid cell development, with the highest activity observed in Ly6C$^{high}$ monocytes, a subset of blood myeloid cells that can differentiate into a heterogeneous group of TAMs[1,27]. Typically, CD206$^+$ MHC II$^-$ TAMs show an M2-like phenotype, while CD206$^-$MHC II$^+$ cells are M1-like polarized, and have higher anti-tumor activity[50–52]. Of note, we found a remarkable increase in the percentage of CD11b$^+$ MHC II$^+$ cells in tumors growing in *Camkk2*$^{-/-}$ mice and that a large fraction of this

heterogeneous population included Ly6C$^{high}$ inflammatory monocytes. An increased percentage of CD206$^-$MHC II$^+$ TAMs and monocyte-derived dendritic cells, and a commensurate decrease in the percentage of CD206$^+$ MHC II$^-$ TAMs were also found in tumors of *Camkk2*$^{-/-}$ mice. Overall, these findings indicate that CaMKK2 plays a critical role in the regulation of processes that determine the fate of myeloid-cell progenitors in the tumor microenvironment and that its deletion/inhibition alters this differentiation program to promote the accumulation of more immunogenic myeloid subsets. The selective expression of CaMKK2 in the myeloid subsets makes this protein an attractive immunotherapeutic target whose inhibition is expected to remodel the tumor microenvironment and stimulate anti-tumor immune responses.

Tumors have a remarkable capacity to modulate the development of myeloid progenitors through their ability to secrete factors that act distally on the bone marrow and splenic myeloid progenitors, or proximally on monocytes recruited to the tumor microenvironment[4,53]. Our data indicate that in the presence of tumor-derived factors, more CD206$^+$ MHC II$^-$ and less CD206$^-$ MHC II$^+$ BMDM are generated in vitro from bone marrow progenitor cells but deletion of *Camkk2* in myeloid cells

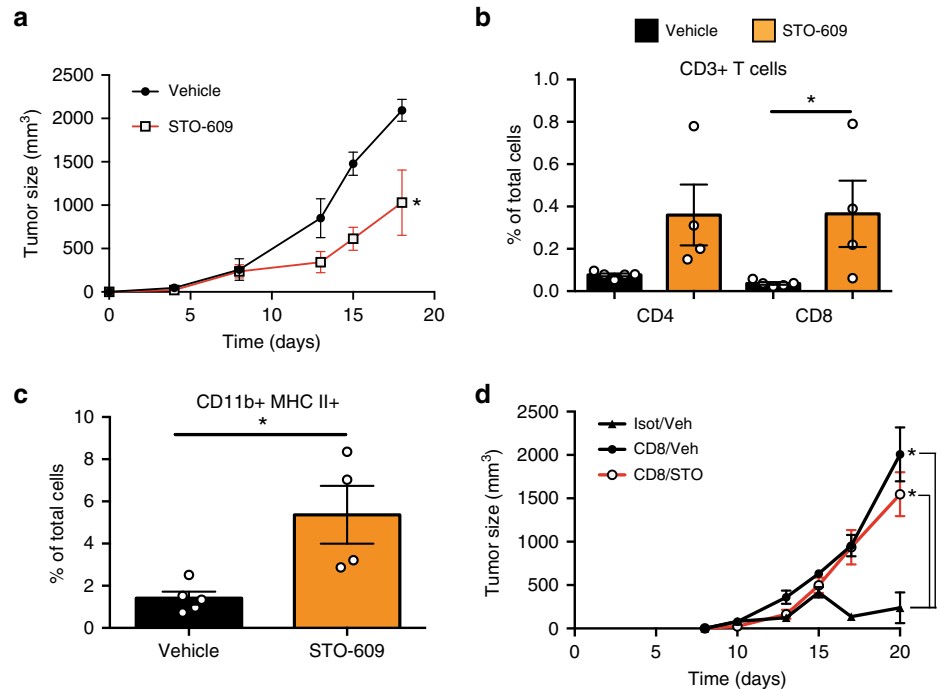

**Fig. 8** Pharmacological inhibition of CaMKK2 attenuates mammary tumor growth in immunocompetent mice. **a** E0771 ($4 \times 10^5$) cells were orthotopically grafted into syngeneic WT mice. Starting at day 2 after grafting, mice were treated three times/week with vehicle or STO-609 (IP, 100 μmoles/kg body weight), and subsequently tumor volumes measured (mean ± SEM; $N = 6$ in each group); two-way ANOVA was used to calculate $p$-values. **b**, **c** T lymphocytes and myeloid cells within E0771 mammary tumors treated with STO-609 or vehicle. Tumors of comparable size (500–700 mm³) were removed, digested and single-cell suspensions were stained for myeloid and lymphoid markers, and analyzed using the gating strategy reported in Supplementary Fig. 3C and 11. Treatment with STO-609 resulted in the accumulation of CD8+ T cells and CD11b+ MHC II+ myeloid cells (b and c, respectively). Bar graph shows mean ± SEM; $N = 5$ and 4 tumors in Veh and STO-609 groups, respectively. A $t$ test was used to calculate $p$-values. **d** STO-609 failed to affect mammary tumor growth in CD8+ T cell-depleted $Camkk2^{-/-}$ mice. $Camkk2^{-/-}$ mice were treated with anti-CD8 or control isotype antibodies. Subsequently, E0771 cells were orthotopically grafted, and mice treated with STO-609 or vehicle. Tumor volumes were measured (mean ± SEM; $N = 3$, 7, and 8 in isotype/Veh, CD8/Veh and CD8/STO groups, respectively). Two-way ANOVA test was used to calculate $p$-values. Asterisks refer to *$p < 0.05$

attenuates this process, leading to the generation of more M1-like and less M2-like macrophages compared with WT myeloid cells. Significantly, $Camkk2^{-/-}$ TCM-BMDM has less capability to suppress T cell proliferation compared with WT TCM-BMDM. Our findings further demonstrated that deletion of $Camkk2$ in BMDM also results in increased expression of genes encoding CXCL9 and CXCL10 chemokines that attract and activate CXCR3 expressing cells, including Th1, CD8+ T cells, and NK cells, which positively regulate anti-tumor immunity[54–59]. The functional relevance of these changes was confirmed using a microfluidic platform that recreates the simulacra of tumors growing in WT and $Camkk2^{-/-}$ mice[48]. These data, in conjunction with more classical measurements of T cell migration, clearly show that more activated CD8+ T cells migrate toward tumor compartments containing $Camkk2^{-/-}$ macrophages.

Deletion of $Camkk2$ facilitates a significant increase in the expression of genes associated with interferon responses (e.g., $Cxcl10$) and with an M1-like phenotype (e.g., $Nos2$ and $CD86$). Besides its effect on chemokines and cytokine gene expression, genetic deletion of $Camkk2$ in BMDM is also associated with decreased expression of several genes that regulate cholesterol biosynthesis/homeostasis ($Hsd17b7$, $Dhcr24$, $Sqle$, $Cyp51$, and $Msmo1$). This is interesting in light of our previously published work demonstrating that mammary tumor growth was dramatically attenuated by inhibition of the CYP27A1-dependent production of the cholesterol metabolite 27-hydroxycholesterol[60,61]. Defining the specific links between CaMKK2, cholesterol, and tumor biology is a current research priority. Finally, despite the finding that CaMKK2 depletion has a positive impact on BMDM

(e.g., enhanced T cell attracting ability and reduced capability to inhibit T cell proliferation), other myeloid cells within TME also express CaMKK2 and thus may also contribute to the tumor growth inhibition observed in CaMKK2-ablated hosts.

It has been suggested by others that inhibition of AMPK could be a useful approach to regulate the immunogenic functions of macrophages and dendritic cells[44,46,62,63]. However, the broad tissue distribution of AMPK, the pleiotropic functions regulated by this kinase in the whole organism and, most importantly, the detrimental effects of AMPK deficiency in CD8+ effector T cells limit the therapeutic utility of drugs that target this enzyme directly[64]. We propose that inhibition of CaMKK2 is a more selective way of inhibiting the fraction of AMPK involved in macrophage polarization while preserving the anti-tumor functions of CD8+ T cells. Thus, notwithstanding any beneficial effects that may be attributed to the inhibition of CaMKK2 within tumor cells, it is clear that this enzyme is also an important myeloid cell-selective immune checkpoint, the inhibition of which may have broad utility in the immunotherapy of breast and other cancers.

## Methods

All of the studies involving the use of animals were conducted after prior approval by the Duke or University of Illinois at Urbana-Champaign Institutional Animal Care and Use Committee (IACUC).

**Mice.** The Tg($Camkk2$-EGFP)DF129Gsat reporter mice were originally provided by the Mutant Mouse Regional Resource Center (MMRRC)[26]. This transgenic strain contains the coding sequence for enhanced green fluorescent protein (EGFP), followed by a polyadenylation signal, inserted into the mouse genomic

bacterial artificial chromosome RP23-31J24 at the ATG transcription initiation codon of the calcium/calmodulin-dependent protein kinase kinase 2 (β) (Camkk2) gene so that expression of the reporter mRNA/protein is driven by the regulatory sequences of the mouse gene. Tg(Camkk2-EGFP)BL/6Gsat mice were generated by backcrossing Tg(Camkk2-EGFP)DF129gsat mice to C57BL/6 mice for ten generations. C57BL/6 mice were purchased from the Jackson Laboratory (CA, USA). Wild-type (WT) and Camkk2[−/−] mice have been described previously[65]. LysMCre + Camkk2[fl/fl] and littermate control LysMCre + Camkk2[wt/wt] mice were generated by crossing B6.129P2-Lyz2[tm1(cre)Ifo/J] mice from the Jackson Laboratory with Camkk2[loxp] mice[65]. Mice were backcrossed to C57BL/6 at least four times during their generation. Genotypes were confirmed by PCR. OTI-EGFP mice were generated by crossing C57BL/6-Tg(TcraTcrb)1100Mjb/J with C57BL/6-Tg(CAG-EGFP)1Osb/J (both strains were acquired from The Jackson Laboratories, Bar Harbor, ME USA). All the mice were used between 8–16 weeks, with gender and age-matched mice used in experimental and control groups. The animals were housed in Duke University animal facilities under a 12-h light/12-h dark cycle with food and water ad libitum. In all, 8–12 week female mice were included in the experiments.

**Human tissues.** We used tissue microarrays (TMA) that included duplicate 1 mm cores of formalin-fixed, paraffin-embedded primary human breast carcinomas from a group of 47 interpretable tumors from Vienna, Austria[66], and 68 samples from Roswell Park Cancer Institute (RPCI), NY[67]. For all tumors, grade and ER/PR/HER2 biomarker data were available. Samples were acquired in compliance with ethical regulations for work with human participant and informed consent policy approved by the corresponding Institutional Review Boards (General Hospital Wiener Neustadt, Austria for the Vienna data set and the Roswell Park Cancer Institute for the RPCI data set). Patient consents were obtained before the collection of tissues.

**CaMKK2 IHC analysis.** TMA sections were deparaffinized, treated with sub-boiling antigen retrieval buffer (citrate, pH 6, Abcam, ab93678) for 20 min, and then reacted with previously validated anti-CaMKK2 rabbit monoclonal antibody[18,49] (HPA017389, Sigma) at 1:500 for 2 h. The detection reaction utilized the rabbit Envision kit from Dako (DakoCytomation, cat. no. K4004). Diaminobenzidine (DAB) was used as the chromogen, with hematoxylin as the counterstain. The IHC experiments were performed on an automated immunostainer (Intellipath from Biocare). Paraffin-embedded blocks of THP-1 cells served as external positive controls. Positive cells showed granular cytoplasmic reactivity. Two board certified pathologists (J.G. and A.H.) performed all analyses, including cell-type identification and staining intensity. Macrophages, endothelial cells, and lymphocytes were identified by morphology. Staining intensity in tumor cells was scored as 0 (absent), 0.5 (borderline), 1 (weak), 2 (moderate), or 3 (strong). For statistical analysis, the tumors were categorized as weak (0, 0.5, 1) or overexpressed (≥2). Ordinal logistic regression was used for binary outcomes, and proportional odds regression was used for grade and molecular class (TN, LA, and LB). There were only a small number of HER2-positive breast cancers in the two cohorts, which were excluded from this analysis. Score was modeled as a binary predictor with levels weak or overexpressed. For outcomes with low cell counts, a Fisher's exact test of association was used. Analyses were conducted in SAS version 9.3 (SAS Institute, Cary, NC) and the R environment for statistical computing.

**Murine mammary tumors.** MMTV-PyMT grafts: MMTV-PyMT mice were backcrossed onto a C57BL/6 background[68]. Resulting mammary tumors were excised, washed in PBS, and diced into equal sized pieces. Tumor pieces were orthotopically grafted into the mammary fat pads of WT or Camkk2[−/−] mice. Resulting tumor growth was assessed by direct caliper measurement.

E0771, 4T1, and Met1 grafts: E0771 and 4T1 cells were obtained from Mark Dewhirst (Duke University) and grown in the RPMI and Dulbecco's modified Eagle Medium (DMEM), respectively, and supplemented with 8% FBS, 0.1 mM nonessential amino acids, 1 mM sodium pyruvate, and penicillin/streptomycin. Met1 cells were obtained from Alexander Borowsky (University of California at Davis) and grown in the DMEM with the same supplements. All cell lines were routinely tested and were free of mycoplasma. None of the cell lines used in this study were found in the database of commonly misidentified cell lines that is maintained by ICLAC and NCBI Biosample. E0771 (200,000), 4T1 (400,000), and Met1 (500,000) cells were grafted orthotopically into a mouse mammary fat pad. 4T1 and Met1 cells were grafted in 50% matrigel. Resulting tumors were assessed by direct caliper measurement. To obtain sizable tumors in Camkk2[−/−] mice for IHC and flow-cytometry studies, a higher number of E0771 cells (4 × 10[5]) were engrafted. This information is reported in the corresponding figure legends.

**Drug treatments.** STO-609 and GSK1901320 (referred to as GSKi) were synthesized at the Duke University Small Molecule Synthesis Facility. These drugs were administered daily by intraperitoneal injection (10–100 μmoles kg[−1] body weight). Sterile dimethyl sulfoxide (Sigma, D2438) was used as a vehicle. STO-609 treatment was initiated at post-graft day 1 for E0771, day 7 for Met1, and day 2 for 4T1 cells. The E0771 tumor growth experiment was repeated independently by a

different investigator using a different batch of STO-609 (Tocris Biosciences, Bristol, UK). Treatment with GSKi was initiated at post-graft day 2.

**Murine tumor dissection.** Tumors were removed from mice and minced with a surgical scalpel. Minced tissues were added to 5 ml of HBSS containing Ca[2+] and Mg[2+] (ThermoFisher Scientific, MA, USA), and supplied with collagenase 2 mg ml[−1] and DNase I 0.1 mg ml[−1] (Roche, Basel, Switzerland). Subsequently, they were transferred to gentle MACS C tubes, dissected with gentle MACS dissociator (Miltenyi, Bergisch Gladbach, Germany), and incubated for 45 min at 37 °C. At the end of this incubation, tumors were dissected again with gentle MACS dissociator, and cell suspensions passed through 70-μm cell strainers (Corning, NY, USA), washed twice in PBS w/2% FBS, and used for FACS analysis.

**Flow cytometry.** Cell suspension (5 × 10[6] cells) from dissected tumors, or BMDM (1 × 10[6]), were stained, with antibodies and fixable viability dye listed in Supplementary Table 2, according to the manufacturer's instructions. The stained cells were analyzed using a BD FACS Canto flow cytometer or BD LSRII (BD, NJ, USA). The data were analyzed using FlowJo (TreeStar, OR, USA).

**CD8[+] cell depletion.** To deplete CD8[+] cells, mice were injected intraperitoneally with 200 μl of anti-mouse CD8 monoclonal antibody (YTS 169.4) or its IgG control isotype (clone LTF-2; catalog #: BE0090, BioXCell, NH, USA) each at a concentration of 1 mg ml[−1] according with regime shown in Supplementary Fig. 4A (the panel S4A was created by L.R.). The effect of CD8 depletion was monitored by flow cytometry using 50 μl of peripheral blood and stained with APC anti-mouse CD8 (6F10) and APC-Cy7 anti-mouse CD45.2 (clone 104; BioLegend).

**L929-differentiation medium and E0771 tumor-conditioned medium (TCM).** L929 cells were acquired from Duke Cell Culture Facility and cultured in the DMEM (Gibco, MA, USA) supplied with 2 mM L-glutamine, 1.0 mM sodium pyruvate (all from Gibco, MA, USA), and 10% fetal bovine serum (FBS, Hyclone, MA, USA). Cells were maintained in a humidified 37 °C CO2 incubator and passed at 70% confluence. To generate conditioned medium, media was removed from 70% confluent cultures and replaced with fresh media. After 3 days, culture supernates were collected and aliquots stored at −80 °C.

E0771 mouse breast cancer cells were cultured in RPMI-1640 (Gibco, MA, USA) supplied with 2 mM L-glutamine, 1.5 g L[−1] sodium bicarbonate, 1.0 mM sodium pyruvate (all from Gibco, MA, USA), and 8% fetal bovine serum (FBS, Hyclone, MA, USA). Cells were maintained in a humidified 37 °C CO2 incubator and passed at 70% confluence. To generate conditioned medium, media was removed from 70% confluent cultures and replaced with fresh media. After 3 days, culture supernates were collected and aliquots stored at −80 °C. Cytokines in conditioned media were analyzed as described below.

**Bone marrow-derived macrophages (BMDM).** To generate BMDM, hind-limb bones were removed from mice and crushed in a mortar with 5 ml of Hanks' balanced salt solution (Mediatech, Manassas, VA) plus 2% FBS (Gemini Bio-Products, West Sacramento, CA) supplemented with 2 mM EDTA. Bone marrow (BM) cell suspensions were passed through a 70-μm strainer (BD Falcon) and stratified on Lympholyte (Cedarlane, Burlington, NC). The low-density fraction containing BM nucleated cells was collected, and the concentration was adjusted to 2 × 10[6] cells ml[−1] in complete medium (DMEM with high glucose and no phenol-red supplemented with 10% fetal bovine serum, glutamine, pyruvate, and HEPES) containing 30% L929-differentiation medium. Subsequently, BM cells were cultured for 5 days in Corning® Costar® ultra-low attachment multi-well plates (Sigma)[24]. To establish the purity of BMDM, cells were double-stained with anti-CD11b and anti-F4/80 antibodies. Based on this analysis, more than 95% of cells co-expressed these markers, which is a characteristic of macrophages. TCM, or an equivalent volume of regular medium (RM), was also added to the differentiation medium in the same experiments as indicated in figure legends. Regular medium was made by RPMI-1640 (Gibco, MA, USA) supplied with 2 mM L-glutamine, 1.5 g L[−1] sodium bicarbonate, 1.0 mM sodium pyruvate (all from Gibco, MA, USA), and 8% fetal bovine serum (FBS, Hyclone, MA, USA).

**Microarray analysis.** WT and Camkk2[−/−] BMDM were generated with L929-conditioned medium (30%) in the presence of RM or TCM (25%). Four biological replicates for each genotype were generated using long bones of 16 WT and 16 Camkk2[−/−] female mice (10–12 weeks old). Microarray analysis was performed at Sequencing and Genome Technologies Shared Resource (Duke University). Transcriptome Analysis Console (TAC) Software (TAC Software version 4.0; Affymatrix) was used for quality control of microarray data and identification of differentially expressed genes (DEGs). iPathwayGuide software (Advatia Bioinformatics) was used for Pathway analysis. PathwayGuide scores pathways using the over-representation of differentially expressed (DE) genes in a given pathway and the perturbation of that pathway computed by propagating the measured expression changes across the pathway topology. These aspects are captured by two independent probability values, pORA and pAcc, that are then combined to generate the pathway-specific p-value, that is calculated with Fisher's method with

Bonferroni correction. Microarray data are available at GEO (accession number: GSE106357).

**Gene set enrichment analysis**. Gene set enrichment analysis (GSEA)[39] was applied to our microarray data. All genes were ranked by the fold-change between the *Camkk2* null and WT control samples. Normalized Enrichment Score (NES) and adjusted q-values were computed utilizing the GSEA method, based on 1000 random permutations of the ranked genes. Gene set collections (MSigDB, http://www.broadinstitute.org/gsea/msigdb/) were used to determine enriched pathways.

**Real-time quantitative RT-PCR assay**. RNA was extracted using the RNeasy Mini Kit (Qiagen, Hilden, Germany). Quantitative PCR was performed using iQ SYBR Green Supermix (Bio-Rad, CA, USA) with respective primers and cDNA. The assay was run on the CFX96 Real-Time System (Bio-Rad, CA, USA). The primer sequences are listed in the Supplementary Table 3.

**Immunoblot**. BMDM were washed three times with 2 ml of ice-cold PBS and lysed with 0.15 ml of M-PER mammalian protein extraction reagent with Halt protease and phosphatase inhibitors (Thermo Scientific). Equal amounts of protein per sample/lane were denatured and resolved by SDS-PAGE. Proteins were transferred to Immobilon-FL membranes (Millipore, Billerica, MA), and quantitative immunoblotting was performed using the Odyssey infrared immunoblotting detection system (LI-COR Biosciences, Lincoln, NE). Primary antibodies used were anti-CaMKK2 (6/CaM Kinase, 610544; BD Biosciences; dilution 1:1000); anti-actin (AC-40, catalog # A4700; Sigma-Aldrich; dilution 1:20000); anti-phospho-CaMK1 (Thr[177], PA5-38434 ThermoFisher, dilution 1:500), anti-phospho-AMPKα (Thr[172], 40H9; catalog # 2535; dilution 1:1000) and AMPKα (F6, catalog # 2793; dilution 1:1000) were from Cell Signaling (Danvers, MA). Secondary antibodies used were anti-mouse IgG Alexa Fluor 680 (catalog # A28183; Invitrogen; dilution 1:1000) and anti-rabbit IgG IRDye800-conjugated antibody (catalog # 611-132-002; Rockland Immunochemicals, Gilbertsville, PA; dilution 1:5000). All antibodies were used according to the manufacturer's instructions. Images of uncut and unmanipulated blots are presented in Supplementary Fig. 12.

**T cell functional assays**. T cells were enriched from the spleen of C57BL6/J mice using the Pan T Cell Isolation Kit II (Miltenyi, Bergisch Gladbach, Germany; catalog # 130-095-130). CD4+ and CD8+ T cells were purified by negative selection using EasySep CD4 and CD8 isolation kits (Stemcell Technology, Cambridge; catalog #19752 and 19753, respectively). Flow cytometry was used to establish degree of purity. After two washings in PBS with 2% FBS and 2 mM EDTA, $1 \times 10^7$ T cells were re-suspended in 1 ml of PBS with 2 µM CFSE (CellTrace™ CFSE Cell Proliferation Kit, Invitrogen; catalog # C34554) and incubated at 37 °C, 5% $CO_2$ for 20 min. In all, 35 ml of RPMI with 10% FBS were subsequently added to the cells and incubated for an additional 10 min. Subsequently, T cells were washed twice in PBS with 2% FBS and 2 mM EDTA. Cell numbers were determined and $1 \times 10^5$ CFSE-labeled T cells, and $1 \times 10^3$ BMDM were seeded into flat-bottom 96-well plates (Corning) with 200 µl of complete media in the presence or absence of 10 ng ml⁻¹ purified NA/LE hamster anti-mouse CD3e (clone 145-2C11; BD Biosciences). After 24 h, supernates were collected for cytokine detection. T cell proliferation was measured at 72 h by flow cytometry. This experiment was replicated using purified CD4+ or CD8+ T cells.

WT and *Camkk2*−/− BMDM were generated in the presence of TCM. The immune-modulatory activity of these supernates was tested on CD4+ and CD8+ T cells. To this purpose, $1 \times 10^5$ CFSE-labeled CD4+ or CD8+ T cells were seeded into 96-well plates (Corning) with 200 µl of complete media with an optimal number of Dynabeads™ Mouse T-Activator CD3/CD28 (ThermoFisher; catalog # 11456D; bead/T cell ratio 1:1) in the presence or absence of TCM-BMDM supernates. T cell proliferation was measured at 72 h by flow cytometry.

T cells isolated from lymph nodes of WT mice were activated for 48 days using anti-CD28 (Biolegend, catalog # 102102; 5 µg ml⁻¹) and anti-CD3 (Invitrogen, catalog # 16-0031-85; 2 µg ml⁻¹) antibody prior to the assay. In total, 500,000 activated T cells or CD8+ OTI T cells were loaded into the top chamber of transwell inserts (VWR, catalog # CLS3464-48EA; 5.0-µm pore size). Supernates from WT and *Camkk2*−/− BMDM, generated in the presence of TCM, was added to the bottom well. Plates were incubated at 37 ºC for the times indicated. The contents of the lower chamber were then collected and analyzed by flow cytometry.

**Cytokine detection**. Cytokines were detected in supernates using MILLIPLEX® 32-MAP Mouse Cytokine/Chemokine Magnetic Bead Panel (EMD Millipore, Darmstadt, Germany) and Luminex® (Luminex, TX, USA), at RBL Immunology Unit of the Regional Biocontainment Laboratory at Duke.

**Tumor microenvironment on-a-chip model**. The polydimethylsiloxane (PDMS) microfluidic devices were fabricated at the CNR-IFN facility using standard soft-lithography methods. Details of the design and microfabrication procedures were reported elsewhere[48]. The PDMS device was composed by three contiguous micro-compartments, including two three-dimensional (3D) lateral tumor micro-environment chambers and a central immune chamber. The lateral chambers were

in continuous contact with medium chambers to guarantee a sufficient exchange of nutrients. Lateral compartments (section $150 \times 500$ µm, length 10,000 µm) were connected with the central compartment (section $150 \times 1200$ µm, length 10,000 µm) via an array of micro-channels (section $10 \times 10$ µm, length 200 µm).

E0771 tumor cells ($1 \times 10^6$ cells) were pre-incubated for 1 h with OVA$_{257-264}$ peptide (1 µM; Sigma-Aldrich), washed and then mixed with BMDM from WT or *Camkk2*−/− mice (E0771/BMDM ratio 1:5). E0771-BMDM cell suspensions were then included in hydrogels of bovine collagen I (PureCol, Advanced Matrix, #5005) at a final concentration of 2.3 mg ml⁻¹, in MEM medium (Sigma-Aldrich), containing 0.28% $NaHCO_3$ (pH = 8) to adjust the final pH to 7. E0771-WT BMDM and E0771 *Camkk2*−/− BMDM were loaded in the right and left compartments, to generate simulacra of 3D WT and *Camkk2*−/− tumor microenvironments (right and left compartment, respectively).

Central compartment was pre-coated with fibronectin (1 µg/ml; Sigma-Aldrich). OVA-specific CD8+ GFP-T cells (OT-I GFP) were isolated from spleen of OT-1 GFP mouse and expanded in vitro for 5 days with OVA$_{257-264}$ peptide and IL-2 (30 U ml⁻¹ R&D System). Bone marrow-derived dendritic cells (BMDC) were generated from Tg(CAG-DsRed*MST)1Nagy/J (The Jackson Laboratory) according to the protocol described elsewhere[69]. BMDC-DsRed were first pre-incubated with OVA$_{257-264}$ peptide for 1 h, washed and then mixed with OT-I-GFP T cells (T cells/BMDC ratio 10:1). The immune cells suspension was finally loaded in the fibronectin-coated central compartment. Microfluidic device was then incubated at 37 °C, 5% $CO_2$ and under humidity control conditions.

Images from 3D-tumor microenvironment compartments (WT and *Camkk2*−/− compartments; see Supplementary Fig. 9) and the central immune compartment were acquired with an Olympus BX51WI upright microscope (Olympus America Inc., Center Valley, PA) using brightfield, FITC and TRITC channel filters, with ×5 objective. Images were taken at time 0 (~1 h after seeding of cells in lymphoid compartment), 18 h, 44 h and 72 h of incubation. Image analysis was performed using Fiji software[69], merging the three channels and performing automated cell count in each fluorescence channel considering equivalent areas in WT and KO compartments.

**Mouse mammary tumors immunocytochemistry**. For immunohistochemistry, tumor sections were deparaffinized and subjected to heat-induced antigen retrieval by incubation in Citrate Buffer pH 6.0 (Abcam, ab93678; Cambridge, MA, USA) for 40 min in a steamer (IHC-Tek™ Epitope Retrieval Steamer, IHC World, LLC, Woodstock, MD, USA). After incubation with 10% normal goat serum for 1 h, the sections were exposed sequentially to anti-CD3 antibody (Abcam; catalog # ab5690; dilution 1:500) or anti-CD31 antibody (Abcam; catalog # ab134168; dilution 1:150) at 4 °C overnight. Sections were then washed and incubated with biotinylated goat anti-rabbit IgG (Vector Laboratories, Burlingame, CA, USA; catalog # BA-1000; dilution 1:200) for 1 h, and subsequently with avidin–biotin-horseradish peroxidase (Vectastain® Elite ABC kit; Vector Laboratories; Burlingame, CA; catalog # PK-4010) for 1 h. Color was developed using 3,3′-diaminobenzidine (DAB) substrate kit (Vector Laboratories, catalog # SK-4100). To identify macrophages, sections were first treated with citrisolv (Fisher Scientific; catalog # 22-143-975) for $2 \times 5$ min; rehydrated 5 min each with 100%, 95%, 80%, 70%, 50%, 20% ethanol and finally water. Sections were then incubated with 0.3% $H_2O_2$ (Sigma, catalog # H1009) in water for 30 min. Sections were then incubated with blocking solution [ABC vectra stain kit (Vector Laboratories; catalog # PK-6200) containing 10% of normal goat serum (Abcam; catalog # ab7481) in PBS-0.3% TWEEN® 20 (Sigma, catalog # P1379) for 20 min. Sections were incubated with F4/80 antibody (Bio-Rad; catalog # MCA497GA; dilution 1:200) at 4 °C overnight, washed, and incubated at room temperature for further 30 min with biotinylated goat anti-rat IgG (Vector Laboratories, Burlingame, CA, USA; catalog # BA-9400; dilution 1:600). Subsequently sections were incubated with avidin–biotin-horseradish peroxidase (Vectastain® Elite ABC kit; Vector Laboratories; catalog # PK-4010) for 1 h. Color was developed using 3,3′-diaminobenzidine (DAB) substrate kit (Vector Laboratories; catalog # SK-4100). Hematoxylin and eosin Stain (Vector Laboratories; catalog # H-3502) and Trichrome Stain (Electron Microscopy Sciences, Hatfield, PA, USA; catalog # 26367) were used according to the recommended protocols. Images were acquired using Olympus BX51WI upright microscope using a ×4, ×20, or ×40 objective. Images were quantified with ITCN plug-in for the ImageJ software.

**Statistics**. Values were assessed for normality and where appropriate either in-transformed or a non-parametric test was selected. Tests used to calculate *p*-values are reported in the figure legends. Graphpad Prism was used for analysis unless otherwise stated. Statistical approaches for Table 1 and Supplementary Table 1 have been described under CaMKK2 IHC Analysis. For the tumor studies, reference data set previously generated in our lab was used to model the experiments, and an endpoint tumor size of 1.2 cm³ was selected. Based on our preliminary data, we assumed an expected difference between groups of 50% between days 11 to 14. Furthermore, we assumed most if not all tumors will reach the selected endpoint. This gives us an expected hazard ratio of 0.36 and setting a type 1 error rate of 0.05 for a 80% power. For power calculation in drug treatment of mammary tumor experiments, we assumed 50% difference between vehicle control and 100 µmoles/kg STO-609. Mice were removed from the study as their tumors reached humane endpoints (2000 mm³). When applicable, randomization to select control and

treated mice/samples and single-blind assessment of the experimental results were used.

**Reporting summary**. Further information on research design is available in the Nature Research Reporting Summary linked to this article.

## Data availability

The Microarray data have been deposited in the GEO database under the accession code GSE106357. All the other data supporting the findings of this study are available within the article and its supplementary information files and from the corresponding authors [LR, DPM] upon reasonable request.

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

## Acknowledgements

This work was supported by a Susan G. Komen grant (IIR13264868) (D.P.M. and L.R.), US Army Medical Research and Material Command grant DOD-W81XWH-15-1-0443 (NC and LR), and NIH grants R01CA174643 (D.P.M.) and R00CA172357 (E.R.N.).

## Author contributions

L.R., E.R.N., W.H., L.B., N.C., D.M., C.y.C. and D.P.M. conceived and designed the experiments; L.R., E.R.N., W.H., S.A.L., W.L., Y.J., S.P., A.E.B., D.M., Y.L., A.M.M. and F.R.B. carried out the experiments; D.H.D., W.J.Z., A.R.M. and B.Y. developed and characterized CaMKK2 inhibitors; A.M.M. carried out the bioinformatics analysis; J.G. and A.H. performed analysis on IHC; L.R., E.R.N., D.M., C.y.C., N.C., A.R.M. and D.P.M. analyzed and interpreted the data; L.R., D.M., C.y.C. and D.P.M. wrote the paper.

## Additional information

**Competing interests:** The authors declare the following competing interests: L.R., E.R.N., W.H., N.C. and D.P.M. have applied for a patent covering the use of CaMKK2 inhibitors alone or in combination with immunotherapy for the treatment of cancer. Title: "CaMKK2 inhibitor compositions and methods of using the same. Racioppi, L., Nelson, E.R., Huang, W., Chao, N. and McDonnell, D.P. Provisional Patent Application No.: 62/371,309; August 5, 2016. The remaining authors declare no competing financial interests.

