## [Peer Review File · Nature Communications]

Reviewers' Comments:

Reviewer #1:

Remarks to the Author:

Authors show a detailed analysis of CaMKK2 and effects in myeloid cells in a tg mice model. Interestingly they show expression of this molecule in tumors as well as the immune TME. Data is lacking to show clear link as well as convincing T cell activation

Qu:

What do they mean that E0771 is a well validated model of luminal BC. Reference 22 seems a weird citation for this. Do they mean that the model is breast cancer or a specific subtype such as estrogen receptor positive which is often referred to as luminal?

Data using human TMAs samples is unclear. Can the authors provide further examples of staining, positive and negative controls? What is high vs low staining? Table 1, the percentages should be by column add to 100%?

Regarding CaMKK2^{-/-} mice and CD3⁺ increases, please show also markers of proliferation and function (i.e. IFNG) on the CD8⁺ and what happens to the CD4⁺? CD8 depletions are interesting but how does myeloid changes affect efficacy in a T cell dependent manner? This link is not clear to me with the data presented. Also changes are shown about MHC II what about MHC I?

Regarding pharmacological inhibition, I assume there is downregulation in the tumor as well as in the stromal immune cells? Can this be shown? Can the authors show more convincing evidence of T cell activation/proliferation and function as well as shift to the M1 phenotype? Where is expression of MHC II and associated ligands? What using their other mouse model (MMTV-PyMT) to be more convincing?

Reviewer #2:

Remarks to the Author:

Racioppi et al describe CaMKK2 as a regulator of the protumoral tumor-associated macrophage phenotype (although nearly all studies were performed on bone marrow-derived macrophages) and, consequently, as a stimulator of tumor progression. Some major issues remain:

1) As a general remark, I find the immunological analyses rather poor when it comes to analyzing the tumor immune infiltrate, the activation state of macrophages and their effect on T-cell activation. Details are outlined below.

2) A second important issue is the use of BMDM (with or without cancer cell conditioned medium, which is still a rather moderate estimate of the real in vivo situation) throughout the study, without assessing the relevance for true tumor-associated macrophages.

3) The gating in Fig1A should be ameliorated. Dendritic cells are not all CD11b(hi). The authors should gate on CD11c(hi)/MHC-II(hi) cells and then subdivide in cDC1 (CD11b-low, CD24-hi), cDC2 (CD11b-hi, Ly6C-lo, CD64-lo) and monocyte-derived DC (CD11b-hi, Ly6C-hi, CD64-hi). Ly6G-hi cells are for sure neutrophils. Eosinophils should be characterized by a high SiglecF expression and high SSC.

Fig 3C: Same comments on the gating as before. This should be much more precise and the authors should put more effort (= more stainings) to correctly identify the subsets of myeloid cells.

4) Fig 2D. The number of endothelial cells is not always informative. The authors should rather assess the functionality of the vessels, including pericyte coverage and perfusion. Also, a quantification must be shown, not merely a "representative" picture.

5) Figs 2E-F. In immunohistochemistry, CD3 and F4/80 are shown. Again, CD3 is not very

informative, as these cells could be CD4+, CD8+, Treg, NKT cells. Co-staining of F4/80 with a marker for profibrotic, restorative macrophages (eg CD206) would also be more informative.

6) Via FACS, the % GzmB+ within CD8+ T cells is shown, but what about the % CD8+ T cells within the hematopoietic compartment or within the viable tumor cell population?

7) On several occasions (eg Fig S1C), the numbers on the figures are hardly readable.

8) Fig 3C-D. MHC-II(hi) macrophages accumulate in KO tumors. Is this also true in similarly sized tumors? Besides comparing WT and Camkk2-KO mice, you also compare large tumors with small tumors. Tumor size, irrespective of genetic background, also influences the nature of the myeloid compartment since smaller tumors typically contain less hypoxic regions.

9) Page 7. One REF is lacking

10) For the in vitro BMDM generation, two distinct set-ups would be required: 1) addition of CM during the differentiation protocol. By doing this, one also investigates the effect of CM on the differentiation process; 2) addition of CM after the BMDM have been generated. This way, the effect of CM on mature macrophages is investigated.

Information should be given whether Camkk2-deficiency as such influences monocyte-to-macrophage differentiation.

11) M1/DC is not a generally accepted term. What are these cells?

12) Page 11. "BMDM conditioned with tumor cell antigens". This is an error.

13) Fig 4D. This panel of genes is far too limited to make a strong claim. In addition, Nos2 gene expression is not always representative of its enzymatic activity in the cells and is counterbalanced by arginase-1, which has not been tested here.

14) Importantly, are the same differences in gene expression, pathways etc found in isolated TAMs from tumors grown in LysM-cre x CaMKK2-floxed versus littermate controls?

15) Figs 5C-D. The situation is more complex than suggested by the authors. While CM increases AMPK phosphorylation in WT cells, it significantly decreases AMPK phosphorylation in KO cells. This means that CM can influence AMPK activity in a CaMKK2-independent fashion. The authors should provide an explanation.

16) The immunological analysis should be strongly improved at several levels: (i) activation state of the macrophages could be investigated to a larger extent, (ii) the activation of T cells should be tested on CD8+ and CD4+ T cells separately (CD4 and CD8 T cells require distinct signals for activation), with an analysis of the Th cytokine/transcription factor profile (Th1, Th2, Th17, Treg)

17) Only testing MHC-II, CD86 and CD40 is really too limited. CD40 is not directly implicated in the stimulation of T cells, while CD80 is crucial, but is amazingly lacking from the analysis.

18) Fig 6A. The authors express this as % MHC-II(high) cells within the F4/80+ population, without defining what they mean by MHC-II(high). Is this population heterogeneous, with a low and a high-expressing population? Does the delta-MFI change in the MHC-II(hi) population? Important is also the use of delta-MFI instead of MFI (MFI marker - MFI isotype control staining), to take into account the level of background staining.

19) Figs 6C-D. Only shown for CD8+ T cells. What about CD4+ T cells and their Th phenotype?

20) Again, can these BMDM findings be recapitulated with isolated TAM from tumors grown in LysM-cre x CaMKK2-floxed versus littermate controls?

21) To prove the absence of off-target effects in the pharmacological studies, the inhibitors should be administered to LysM-cre x CaMKK2-floxed, in which they should have no additional effect

Reviewer #3:

Remarks to the Author:

In this manuscript by Racioppi, et al. the authors demonstrate that the absence of Camkk2 in myeloid cells significantly reduces the growth of transplanted luminal B mammary carcinomas, in a manner that depends upon increased infiltration/effector function of cytotoxic CD8+ T cells. Similar results are obtained with an inhibitor of CaMKK2, suggesting this may represent a therapeutic target. Mechanistically the authors ascribe their phenotype to changes in the number of MHC II+ macrophages and/or a reduced ability of bone marrow-derived 'tumor' macrophages (BMDMs) to mediate suppression in vitro. However, the authors do not demonstrate that these changes in

macrophages are functionally relevant, and the lack of in vivo analysis and validation significantly diminishes the impact of the manuscript.

Major Points:

1. The manuscript nicely uses two orthotopic models of luminal B-like mammary carcinoma to demonstrate that Camkk2 deficiency reduces/prevents tumor growth. However, the therapeutic relevance of this is not properly tested in Figure 7/S6 as the animals are treated only 2 days following tumor implantation. To demonstrate therapeutic targeting, tumors need to be allowed to grow to at least 100 mm³ prior to treatment initiation. Extending these findings to a transgenic model (e.g. MMTV-PyMT) would also be welcome, although not necessary for publication.
2. The authors need to show that the altered phenotype of Camkk2-deficient BMDMs in vitro corresponds to a change in macrophage phenotype in vivo.
3. There is no demonstration that these changes are responsible for the reduced tumor growth. For example, macrophages appear to be a minor subset of myeloid cells within the E0771 tumors, in contrast to the prominence of this population within PyMT tumors. Camkk2 expression is observed in the various myeloid subsets, and therefore these may be responsible for the reduction in tumor growth shown using the LysMCre model, while the increase in MHCII⁺ macrophages may simply reflect reduced tumor growth and/or increased level of cell death.
4. Similarly, the authors describe global gene expression changes in BMDMs, but do not link a specific pathway to changes in the ability of BMDMs to suppress T cell proliferation (e.g. Arg1, Nos2). Without additional mechanistic insight the impact of the manuscript is minimal.
5. In regards to the clinical application, the authors stated that CaMKK2 is overexpressed in triple negative (TN) breast cancer; however, this is not apparent from the table presented, as they only show correlation with ER/PR/Her2 status (and not all 3). Furthermore, the authors do not evaluate the impact of Camkk2 in a model of TNBC/basal disease. Camkk2 expression in the cell lines used is not described, but would high expression within tumor cells potentially alter the response observed?
6. The manuscript contains numerous errors, missing references, and residual tracked changes. Some of these have been noted below.

Minor issues:

1. The authors should include which statistical test was used to determine significance in each experiment in the figure legend. The number of experimental repeats, number of replicates and whether the replicates are biological or technical should also be included in the legend for each figure and subfigure.
2. An expanded explanation of CaMKK2 and the cellular CaM kinase cascade in the introduction would be welcome. In particular the signaling pathways and transcription factors regulated by these proteins.
3. The analysis of FSC to measure increased T cell activation is limited, and there appear to be only 3 tumors analyzed. The PD-1/Gzmb analysis has not been quantified, and again appears to be from only 3 tumors.
4. Figure 2A is missing the statistics/p-value.
5. Quantification of the CD31 staining shown in figure 2D should be performed to rule out differences. This should include an analysis of vessel size.
6. In figure 2E one of the images is missing a scale bar and appears to be of higher magnification.
7. The CD8 clone used throughout the paper is not indicated in the methods section. The CD8 separation in figure S1A is minimal, whereas there is decent separation in figure S7. Was the same clone and conjugate used for both?
8. In figure S2A-B, what is meant by the y-axis label [CD8 (% of Ly)]?
9. In figure S2D, could the authors provide reasoning for the reduction in CaMKK2 protein levels in the LysM-Cre+Camkk2wt/wt mice as compared to the overall WT mice? It appears that the LysM-Cre may be having an effect on CaMKK2 levels.
10. In figure S3A, the x-axis would be easier to read if the authors put the numbers in scientific notation.

11. Figure S3C is potentially helpful in understanding the complex BMDM co-culture experiment, but the labels don't match what's described in the text. A more complete diagram, including all of the conditions used might also make the figure more effective.
12. It's not clear what figure 5A adds to the paper. The fact that CaMKI and AMPK are downstream of and phosphorylated by CaMKK2 has already been published.
13. In figure 5B, in the E0771-conditioned medium samples, the CaMKK2 protein lost the double banding feature observed in the other instances of CaMKK2 western blotting throughout the paper. Is this an artifact of the blot, or does CaMKK2 have an isoform or post-translational modification that the conditioned media is affecting?
14. In Figure 6A the authors switch from showing marker expression as a percentage of F4/80 to showing the MFI. It would be better to be consistent throughout the panel, and possibly to show the alternative analysis in supplemental data.
15. The FSC/SSC gate does not appear accurate in Figure S7. This approach may also miss activated lymphocyte (as the authors showed in Figure S1). Simply gating on CD45, CD3, and then CD4/CD8 would be ideal.
16. It is suggested that the increase in CXCL10 levels is due to T cell expression, but macrophages can also express the chemokine in response to IFN- γ .
17. GFP-negative (background) controls are missing from Figure 1. It is difficult to know whether expression is truly positive in the figure.
18. The method section does not describe the (surgical?) procedure used to engraft tumor pieces into mammary fat pads.
19. Figure S2E is not properly referenced in the figure legend.

Reviewers' comments:

Reviewer #1:(Remarks to the Author):

Authors show a detailed analysis of CaMKK2 and effects in myeloid cells in a tg mice model. Interestingly they show expression of this molecule in tumors as well as the immune TME. Data is lacking to show clear link as well as convincing T cell activation

Qu:

1. What do they mean that E0771 is a well validated model of luminal BC. Reference 22 seems a weird citation for this. Do they mean that the model is breast cancer or a specific subtype such as estrogen receptor positive which is often referred to as luminal?
 - E0771 is a model for ER+ breast cancer. However, in this paper we have not limited our studies to models of one subtype of breast cancer. The wording has been changed to reflect this fact and Ref 25 added (as requested).
2. Data using human TMAs samples is unclear. Can the authors provide further examples of staining, positive and negative controls? **What is high vs low staining? Table 1, the percentages should be by column add to 100%?**
 - We (and others) have previously published the validation data for the antibody used for the IHC analysis (Refs 18 and 48). The manuscript has been revised to include this information.
 - **Examples of high vs low CaMKK2 staining have been provided (Figure S1A).**
 - **In Table 1, percentages by row add to 100% (i.e. 75% and 40% indicates the % of LA in CaMKK2 Low and high groups, respectively). Table 1 has been revised and discussed in the text.**
3. Regarding CaMKK2^{-/-} mice and CD3⁺ increases, please show also markers of proliferation and function (i.e. IFNG) on the CD8⁺ and what happens to the CD4⁺? **CD8 depletions are interesting but how does myeloid changes affect efficacy in a T cell dependent manner? This link is not clear to me with the data presented. Also changes are shown about MHCII what about MHCI?**
 - This has been addressed by including data on the expression of T cell activation and inhibitory markers (GzmB, CD69, and LAG-3; Fig. 3B). No change in the total CD4 population was observed when comparing tumors from WT and CaMKK2^{-/-} mice. We did observe decreased expression of LAG3 and PD1 in CD4⁺ TIL isolated from KO mice (Fig. 3B).
 - **Addressed in the text: Tumor infiltrating myeloid cells are likely to play an important role in regulating T cell function in KO mice. This is supported by the phenotypes of the infiltrating myeloid cells in CaMKK2^{-/-} mice (less M2-like Mac, more M1-like Mac and more monocyte-derived DC cells), one would predict that the T cells within these tumors would be more active in eliciting an anti-tumor response. Significantly, TAM from KO tumors also secrete more chemokines involved in the recruitment of effector T cells. This finding has been replicated in vitro with BMDM. An increased recruitment of T cells in conjunction with less immunosuppressive activity would explain the increased efficacy of anti-tumor immune response in KO mice.**

- Addressed. The levels of MHC I expression on BMDM generated in the presence or absence of TCM were analyzed and shown in the Fig. S8A, lower left panel. Regardless of genotype, increased levels of MHC I were found in TCM-BMDM compared to RM-BMDM. However, lower levels of MHC I were found in Camkk2-/- BMDM generated in either RM or TCM compared to WT. This finding was included in the revised manuscript.
4. Regarding pharmacological inhibition, I assume there is downregulation in the tumor as well as in the stromal immune cells? Can this be shown? **Can the authors show more convincing evidence of T cell activation/proliferation and function as well as shift to the M1 phenotype? Where is expression of MHCII and associated ligands? What using their other mouse model (MMTV-PyMT) to be more convincing?**
- The CaMKK2 inhibitors we describe/use do not downregulate CaMKK2 expression but directly inhibit enzymatic activity. The importance of host (i.e. non-tumor cell) CaMKK2 expression in tumor immunity was confirmed by showing that the myeloid specific deletion of CaMKK2 mirrored the phenotypes observed in the global knockout and that the effect of either manipulation could be reversed by depletion of CD8+ T cells (Fig 7D). We cannot rule out the possibility that inhibition of cancer cell intrinsic CaMKK2 activities also contributes to the phenotypic responses observed with pharmacological inhibitors. Evaluation of the tumor cell intrinsic effects of CaMKK2 is an active area of research in our laboratory although beyond the scope of this manuscript.
 - **Treatment with STO-609 is associated with increased accumulation of T cells and MHC II+ myeloid cells and TILs (Fig 7). These data phenocopy in part the effects of genetic CaMKK2 depletion on tumor immune cell repertoire. Our studies do not exclude potential direct effects of STO-609 on tumor cells although the lack of activity of the CaMKK2 inhibitors upon depletion of CD8+ T cells (Fig. 7D) highlight the importance of immune cells as target of CaMKK2 inhibitors.**
 - We confirmed the effects of STO609 in both the Met1 and 4T1 models. Data for MMTV-PyMT model were also included in the revised manuscript (Fig S10A-C)

Reviewer #2: (Remarks to the Author):

Racioppi et al describe CaMKK2 as a regulator of the protumoral tumor-associated macrophage phenotype (although nearly all studies were performed on bone marrow-derived macrophages) and, consequently, as a stimulator of tumor progression. Some major issues remain:

1) As a general remark, I find the immunological analyses rather poor when it comes to analyzing the tumor immune infiltrate, the activation state of macrophages and their effect on T-cell activation. Details are outlined below.

- We have addressed this issue by dealing with each specific issue in detail (below).

2) A second important issue is the use of BMDM (with or without cancer cell conditioned medium, which is still a rather moderate estimate of the real in vivo situation) throughout the study, without assessing the relevance for true tumor-associated macrophages.

- As requested we extended our studies to evaluate phenotype and gene expression in tumor associated macrophages (Fig 6D-E and S8D).

3) The gating in Fig1A should be ameliorated. Dendritic cells are not all CD11b(hi). The authors should gate on CD11c(hi)/MHC-II(hi) cells and then subdivide in cDC1 (CD11b-low, CD24-hi), cDC2 (CD11b-hi, Ly6C-lo, CD64-lo) and monocyte-derived DC (CD11b-hi, Ly6C-hi, CD64-hi). Ly6G-hi cells are for sure neutrophils. Eosinophils should be characterized by a high SiglecF expression and high SSC.

Fig 3C: Same comments on the gating as before. This should be much more precise and the authors should put more effort (= more staining) to correctly identify the subsets of myeloid cells.

- The gating strategy to identify myeloid cells of interest has been ameliorated adding more staining (Figs. S1B, S3C-D).

4) Fig 2D. The number of endothelial cells is not always informative. The authors should rather assess the functionality of the vessels, including pericyte coverage and perfusion. Also, a quantification must be shown, not merely a “representative” picture.

- Addressed in the text: We cannot completely rule out the possibility that CaMKK2 expressed in myeloid cells may also affect neo-angiogenesis in tumors and have modified the discussion accordingly. However, we have quantified the CD31 staining and see no differences in WT and CaMKK2 KO mice.

5) Figs 2E-F. In immunohistochemistry, CD3 and F4/80 are shown. Again, CD3 is not very informative, as these cells could be CD4+, CD8+, Treg, NKT cells. Co-staining of F4/80 with a marker for profibrotic, restorative macrophages (eg CD206) would also be more informative.

- This has been addressed by including a more extensive analysis of immune cells infiltrating tumors removed from WT and Camkk2 -/- mice. More staining and a previously validated gating strategy (ref. 28) were used to identify immune cell types, including CD206+ restorative macrophages. The expression of immune regulatory genes was also evaluated in tumors from WT and KO mice. These data have been included in Figs. 1E-F; 3A-B; 6D and S8D.

6) Via FACS, the % GzmB+ within CD8+ T cells is shown, but what about the % CD8+ T cells within the hematopoietic compartment or within the viable tumor cell population?

- Addressed in Fig 3.b (new).

7) On several occasions (eg Fig S1C), the numbers on the figures are hardly readable.

- Addressed.

8) Fig 3C-D. MHC-II(hi)macrophages accumulate in KO tumors. Is this also true in similarly sized tumors? Besides comparing WT and Camkk2-KO mice, you also compare large tumors with small tumors. Tumor size, irrespective of genetic background, also influences the nature of the myeloid compartment since smaller tumors typically contain less hypoxic regions.

- In the revised manuscript, datasets obtained using comparable-size tumors has been included (Figs. 2E; 3A-B)

9) Page 7. One REF is lacking

- Addressed

10) For the in vitro BMDM generation, two distinct set-ups would be required: 1) addition of CM during the differentiation protocol. By doing this, one also investigates the effect of CM on the differentiation process; 2) addition of CM after the BMDM have been generated. This way, the effect of CM on mature macrophages is investigated. Information should be given whether Camkk2-deficiency as such influences monocyte-to-macrophage differentiation.

- Addressed: To evaluate the effects of TCM on already differentiated BMDM, cells were first cultured in the presence of RM for 4 days, and then exposed for additional 48 hours to TCM. These experiments confirmed the ability of TCM to increase the percentage of M2-like cells in WT BMDM and importantly also demonstrated that this response was attenuated in Camkk2^{-/-} BMDM (Supplementary Fig 8B). This is also described in the text (Page 15)

11) M1/DC is not a generally accepted term. What are these cells?

- Addressed: In Fig.4, we use M1 & DC to indicate genes expressed in both cell types M1 and DC).

12) Page 11. "BMDM conditioned with tumor cell antigens". This is an error.

- Corrected

13) Fig 4D. This panel of genes is far too limited to make a strong claim. In addition, Nos2 gene expression is not always representative of its enzymatic activity in the cells and is counterbalanced by arginase-1, which has not been tested here.

- We agree with the reviewer's point. A number of the genes associated with M1-like macrophages were induced in KO TCM-BMDM in the microarray analysis (Figs. 4C and S6), although a few genes associated with the M2-like phenotype were also induced (e.g. Arg1). We have discussed this in the text. The molecular basis for this specific gene expression pattern is not yet known but the important finding is that CaMKK2 knockdown in macrophages reprograms them in a manner which facilitates enhanced T cell proliferation and migration (Fig 6 F-I).

14) Importantly, are the same differences in gene expression, pathways etc found in isolated TAMs from tumors grown in LysM-cre x CaMKK2-floxed versus littermate controls?

- We have evaluated gene expression in TAMs from tumors propagated in the WT and global knockout mice but not in tumors from the conditional LysMCre mice. Considering the difficulty obtaining enough TAMs for these types of experiments we request that the reviewer give us a "pass on this one" as it would require extensive breeding on our part to generate the numbers of mice required for the requested experiment.

15) Figs 5C-D. The situation is more complex than suggested by the authors. While CM increases AMPK phosphorylation in WT cells, it significantly decreases AMPK phosphorylation in KO cells. This means that CM can influence AMPK activity in a CaMKK2-independent fashion. The authors should provide an explanation.

- Addressed in text: CaMKK2 is not the only upstream kinase for AMPK. Besides CaMKK2, the best characterized kinase for AMPK is LKB1, which phosphorylates AMPK in response to nutrient deprivation or hypoxia. We have clarified this point in the text.

16) The immunological analysis should be strongly improved at several levels: (i) activation state of the macrophages could be investigated to a larger extent, (ii) the activation of T cells should be tested on CD8+ and CD4+ T cells separately (CD4 and CD8 T cells require distinct signals for activation), with an analysis of the Th cytokine/transcription factor profile (Th1, Th2, Th17, Treg).

- As requested TAMs were identified and stained using a larger panel of markers. In addition, we analyzed gene expression in sorted TAM and determined that the CXCL- chemokines (Cxcl9, Cxcl10 and Cxcl14) were upregulated in TAM of Camkk2 KO. We tested the effects of BMDM supernates on purified CD4 and CD8 T cells proliferation. The conclusion of this study indicated that supernates from WT TCM-BMDM inhibits proliferation of CD8+ T cells more efficiently than supernates from KO TCM-BMDM. Similar data were obtained using intact TCM-BMBM from WT/KO. Although CD4 cells did not proliferate well under these conditions, a similar inhibitory pattern was observed in this subset. Further studies will be required to identify the inhibitory factor(s) released by BMDM-TCM that is/are regulated by CaMKK2. These data have been included in the revised Figs. 2E-F; 3A-B; 6A-E, 6H-I; Figs. S1B; S2; S3B-C; S8 and S9.

17) Only testing MHC-II, CD86 and CD40 is really too limited. CD40 is not directly implicated in the stimulation of T cells, while CD80 is crucial, but is amazingly lacking from the analysis.

- Addressed: The levels of CD80 expression on BMDM generated in the presence or absence of TCM were analyzed and shown in the Fig. S8. Regardless of genotype, increased levels of CD80 were found in TCM-BMDM compared to RM- BMDM. However, lower levels of CD80 were found in Camkk2^{-/-} BMDM generated in either RM or TCM compared to WT. We discussed this finding in the text.

18) Fig 6A. The authors express this as % MHC-II(high) cells within the F4/80+ population, without defining what they mean by MHC-II(high). Is this population heterogeneous, with a low and a high-expressing population? Does the delta-MFI change in the MHC-II(hi) population? Important is also the use of delta-MFI instead of MFI (MFI marker - MFI isotype control staining), to take into account the level of background staining.

- Addressed. The expression of MHC II was reported as MFI in Fig 6.B (new). MFI isotype control was negligible compared to the MHC II expression (MFI control <

100 vs MFI MHC II > 1000). In addition, the levels of the background controls (MFI of the isotype control) were comparable between the different BMDMs preparations (RM vs TCM and WT vs KO). For these reasons, we used MFI to evaluate changes in MHC II expression (Fig. 6B).

19) Figs 6C-D. Only shown for CD8+ T cells. What about CD4+ T cells and their Th phenotype?

- These data are reported in the Figs. S6H-I. The original Fig 6C-D is now Fig 6F-G.

20) Again, can these BMDM findings be recapitulated with isolated TAM from tumors grown in LysM-cre x CaMKK2-floxed versus littermate controls?

- Addressed. We have harvested tumors from WT and Camkk2^{-/-} mice and examined the immune phenotype. The results are consistent with what we see in-vitro with BMDMs (Figs. 6A, C-D).

21) To prove the absence of off-target effects in the pharmacological studies, the inhibitors should be administered to LysM-cre x CaMKK2-floxed, in which they should have no additional effect.

We know that both STO609 and GSKi can inhibit other kinases at higher concentrations and are used in this study to evaluate whether the phenotypes observed in the CaMKK2 KO mice can be recapitulated using "tool" compounds. We have an ongoing program with the structural biology consortium to develop inhibitors with improved pharmaceutical properties. Use of two structurally unrelated compounds, GSKi and STO609, in the study was to ease the concern of off-target effects (Fig S10 D-E).

Reviewer #3 (Remarks to the Author):

In this manuscript by Racioppi, et al. the authors demonstrate that the absence of Camkk2 in myeloid cells significantly reduces the growth of transplanted luminal B mammary carcinomas, in a manner that depends upon increased infiltration/effector function of cytotoxic CD8+ T cells. Similar results are obtained with an inhibitor of CaMKK2, suggesting this may represent a therapeutic target. Mechanistically the authors ascribe their phenotype to changes in the number of MHCII+ macrophages and/or a reduced ability of bone marrow-derived 'tumor' macrophages (BMDMs) to mediate suppression in vitro. However, the authors do not demonstrate that these changes in macrophages are functionally relevant, and the lack of in vivo analysis and validation significantly diminishes the impact of the manuscript.

Major Points:

1. The manuscript nicely uses two orthotopic models of luminal B-like mammary carcinoma to demonstrate that Camkk2 deficiency reduces/prevents tumor growth. However, the therapeutic relevance of this is not properly tested in Figure 7/S6 as the animals are treated only 2 days following tumor implantation. To demonstrate therapeutic targeting, tumors need to be allowed to grow to at least 100 mm³ prior to treatment initiation. Extending these findings to a transgenic model (e.g. MMTV-PyMT) would also be welcome, although not necessary for publication.

- We performed the requested experiment in E0771 tumors and found that administration of the drug when the tumors are this size is ineffective. E0771 is a particularly fast-growing tumor and the delayed administration of the drug may not leave enough time to reprogram the tumor microenvironment in a manner that impacts growth. However, we have included a new set of data showing the efficacy of STO-609 treatment in the established MMTV PyMT model (Fig. S10C).

2. The authors need to show that the altered phenotype of Camkk2-deficient BMDMs in vitro corresponds to a change in macrophage phenotype in vivo.

- Addressed in Figs. 6D, S8C-D (new)

3. There is no demonstration that these changes are responsible for the reduced tumor growth. For example, macrophages appear to be a minor subset of myeloid cells within the E0771 tumors, in contrast to the prominence of this population within PyMT tumors. Camkk2 expression is observed in the various myeloid subsets, and therefore these may be responsible for the reduction in tumor growth shown using the LysMCre model, while the increase in MHCII+ macrophages may simply reflect reduced tumor growth and/or increased level of cell death.

- Addressed in text: We have clarified this point in the text by more appropriately attributing effects noted to inhibition of CaMKK2 in myeloid cells as opposed to macrophages (with the exception of the in vitro studies when we refer to bone marrow derived-macrophages). We have added discussion to clarify this point.

4. Similarly, the authors describe global gene expression changes in BMDMs, but do not link a specific pathway to changes in the ability of BMDMs to suppress T cell proliferation (e.g. Arg1, Nos2). Without additional mechanistic insight the impact of the manuscript is minimal.

- Addressed: We have included the T cell migration data (and CXCL-expression) as a likely mechanism (Figs 6E, G, H; new).

5. In regards to the clinical application, the authors stated that CaMKK2 is overexpressed in triple negative (TN) breast cancer; however, this is not apparent from the table presented, as they only show correlation with ER/PR/Her2 status (and not all 3). Furthermore, the authors do not evaluate the impact of Camkk2 in a model of TNBC/basal disease. Camkk2 expression in the cell lines used is not described, but would high expression within tumor cells potentially alter the response observed?

- Addressed. In the table 1, we show that CaMKK2 expression is negatively associated with LA (in "Vienna" and "combined" samples). On the contrary, CaMKK2 expression is positively associated with TN, defined as ER-, PR-, Her2-, ("RPCI" and "combined"). Based on these data we concluded that CaMKK2 expression correlates negatively with LA, and positively with TN type. These results were discussed in the revised manuscript. Table S2 was removed from the revised supplemental due to the partially redundancy with Table 1.
- 4T1 and Met1 included in Fig. S10 A-B are both ER- PR- models of breast cancer.
- We have detected CamKK2 expression in E0771 cells but have not evaluated the expression of Camkk2 in these other cell lines. We cannot formally rule out the possibility that the expression of Camkk2 in cancer cells may alter the response, however, the importance of host (ie non-tumor cell) CaMKK2 expression in tumor immunity was demonstrated in both myeloid-specific deletion of CaMKK2 and in the global knockout, and that the effect of either manipulations could be reversed by depletion of CD8+ T cells (Fig. 7D).

6. The manuscript contains numerous errors, missing references, and residual tracked changes. Some of these have been noted below.

- Corrected all errors

Minor issues:

1. The authors should include which statistical test was used to determine significance in each experiment in the figure legend. The number of experimental repeats, number of replicates and whether the replicates are biological or technical should also be included in the legend for each figure and subfigure.

- Addressed

2. An expanded explanation of CaMKK2 and the cellular CaM kinase cascade in the introduction would be welcome. In particular the signaling pathways and transcription factors regulated by these proteins.

- Addressed (page 3-4)

3. The analysis of FSC to measure increased T cell activation is limited, and there appear to be only 3 tumors analyzed. The PD-1/Gzmb analysis has not been quantified, and again appears to be from only 3 tumors.

- Addressed: Data in Fig. 3 were confirmed and extended in an additional experiment. As suggested by reviewer, FSC as measure of T cell activation was removed, and the expression of activation/inhibitory molecules (Gzmb, PD-1, CD69 and LAG3) on CD4+ and CD8+ TIL was shown (Figs. 3A-B; new).

4. Figure 2A is missing the statistics/p-value.

- Addressed: ($p < 0.001$)

5. Quantification of the CD31 staining shown in figure 2D should be performed to rule out differences. This should include an analysis of vessel size.

- Figure 2D is now Fig S2B and the quantification of CD31 (expressed as percentage of CD31 stained area/tumor area) is shown below the IHC graph and we see no differences in WT and CaMKK2 KO mice (Fig. S2B). We cannot completely rule out the possibility that CaMKK2 expressed in myeloid cells may also affects neo-angiogenesis in tumors and have modified the discussion accordingly.

6. In figure 2E one of the images is missing a scale bar and appears to be of higher magnification.

- Addressed: the upper left panel has been replaced with one showing the scale bar.

7. The CD8 clone used throughout the paper is not indicated in the methods section. The CD8 separation in figure S1A is minimal, whereas there is decent separation in figure S7. Was the same clone and conjugate used for both?

- Addressed: CD8 clone used for staining has been reported in the new Table S4. The gating strategy used for TIL analysis has been shown in Fig. S3 (new).

8. In figure S2A-B, what is meant by the y-axis label [CD8 (% of Ly)]?

- Addressed: The original Fig. S2B is now Fig. S4B and the Y-axis has been relabeled as % of CD3+.

9. In figure S2D, could the authors provide reasoning for the reduction in CaMKK2 protein levels in the LysM-Cre+Camkk2wt/wt mice as compared to the overall WT mice? It appears that the LysM-Cre may be having an effect on CaMKK2 levels.

- Addressed: In the current version of the manuscript, this data has been included in the Fig. S4 D. As reported in the legend of this figure, WT and global Camkk2-/- BMDM have been used as positive and negative controls for CaMKK2 expression. However, fresh isolated peritoneal macrophages are used to determine CaMKK2 expression in LysMS Cre CaMKK2loxP. Differences in total CaMKK2 expression might relate to cell types (BMDM vs peritoneal macrophages).

10. In figure S3A, the x-axis would be easier to read if the authors put the numbers in scientific notation.

- Addressed: numbers are more readable in the revised figure (Fig. S3A corresponds to Fig. S5A in the revised manuscript).

11. Figure S3C is potentially helpful in understanding the complex BMDM co-culture experiment, but the labels don't match what's described in the text. A more complete diagram, including all of the conditions used might also make the figure more effective.

- This figure is now Fig. S5C. Labels have been revised to match with text.

12. It's not clear what figure 5A adds to the paper. The fact that CaMKI and AMPK are downstream of and phosphorylated by CaMKK2 has already been published.

- Fig. 5A has been removed as requested

13. In figure 5B, in the E0771-conditioned medium samples, the CaMKK2 protein lost the double banding feature observed in the other instances of CaMKK2 western blotting throughout the paper. Is this an artifact of the blot, or does CaMKK2 have an isoform or post-translational modification that the conditioned media is affecting?

- Although we are still investigating this interesting phenomenon, we might reasonable rule out that the collapse of the CaMKK2 doublet in an apparently single band is due to an artifact of the blot. CaMKK2 has multiple isoforms, multiple phosphorylation sites and probably might undergone to post-transcriptional modifications. Further studies are required to verify the molecular bases of CaMKK2 electrophoretic mobility changes in TCM BMDM.

14. In Figure 6A the authors switch from showing marker expression as a percentage of F4/80 to showing the MFI. It would be better to be consistent throughout the panel, and possibly to show the alternative analysis in supplemental data.

- Addressed: In the new Fig. 6, the percentages of, and gating strategy to identify, MHC II+ and MHCII- cells are reported in Fig. 6A and 6D. MHC II expression shown as MFI is also included as Fig 6B.

15. The FSC/SSC gate does not appear accurate in Figure S7. This approach may also miss activated lymphocyte (as the authors showed in Figure S1). Simply gating on CD45, CD3, and then CD4/CD8 would be ideal.

- Addressed: The gating strategy has been revised and shown in Fig. S11A.

16. It is suggested that the increase in CXCL10 levels is due to T cell expression, but macrophages can also express the chemokine in response to IFN- γ .

- Addressed: The reviewer is correct, and it is not possible to identify the source of CXCL10 under such co-culture conditions. For this reason, this data has not been included in the revised manuscript. However, an extensive analysis of chemokine expression in BMDM as well as in TAM isolated from tumors of WT and KO mice have been included (Figs. 6C, E; S8 C-D).

17. GFP-negative (background) controls are missing from Figure 1. It is difficult to know whether expression is truly positive in the figure.

- Addressed: A bar indicating the expression level of GFP negative splenocytes has been included in the Fig. 1C.

18. The method section does not describe the (surgical?) procedure used to engraft tumor pieces into mammary fat pads.

- Addressed: Page 26.

19. Figure S2E is not properly referenced in the figure legend.

- Addressed: This is now Fig. S4D.

Reviewers' Comments:

Reviewer #1:

None

Reviewer #2:

Remarks to the Author:

This manuscript has greatly improved and the authors for the most part responded successfully to the remarks.

A few minor issues remain:

1) The gating in Fig 1SB: Things I don't understand:

- when gating on the Ly6G- population: the CD11c/MHC-II plot does not show a major population with high CD11c expression; the CD11c/CD11b plot shows a major population with very high CD11c expression. This seems impossible since the parent populations for both plots should be the same.

- Monocytes, immature macrophages and macrophages are all gated based on the CD11b/CD11c plot, which contain a very prominent population of CD11c(high) cells. This would mean that all monocytes and macrophages in these tumors are very high for CD11c, which is most likely not the case, and is contradicted by the CD11C/MHC-II plot.

- The Mac gating (Ly6C/F4/80 plot) follows a pre-gating on cells that are mostly MHC-II-positive (in the SSC-A/MHC-II plot). Nevertheless, 34% of these Macs are MHC-II-negative, which is again a very surprising finding. In the SSC-A/MHC-II plot there are indeed also MHC-II-negative cells, but these have a very high SSC, suggesting that these cells are rather eosinophils than macrophages.

2) The authors mention that MoDC can associate with a better CD8+ T-cell response (ref 37). MoDC have also been reported to be mainly immunosuppressive (Laoui et al, Nat Commun, 2016, 7:13720), so this possibility should be included as well.

3) I agree that the KO macrophages are overall more M1-oriented and thus likely more immunogenic. However, a combined upregulation of iNOS and arginase is often associated with the induction of an immunosuppressive program. This should at least be mentioned in the text.

Reviewer #3:

Remarks to the Author:

The authors have adequately addressed all of my concerns.

Response to Reviewers.

Reviewers 1 and 3 were satisfied with the responses we provided to their comments.

Reviewer #2 communicated the following:

This manuscript has greatly improved and the authors, for the most part, responded successfully to the remarks.

A few minor issues remain:

(1) The gating in Fig 1SB: Things I don't understand:

- when gating on the Ly6G- population: the CD11c/MHC-II plot does not show a major population with high CD11c expression; the CD11c/CD11b plot shows a major population with very high CD11c expression. This seems impossible since the parent populations for both plots should be the same.

Response: We thank the referee for this comment. Upon rereading the manuscript, we realized that we had mislabeled the axes in Fig. S1B (CD11c/CD11 labels were inverted). The correct label is "CD11b" on the X-axis and "CD11c" on the Y-axis.

- Monocytes, immature macrophages and macrophages are all gated based on the CD11b/CD11c plot, which contain a very prominent population of CD11c(high) cells. This would mean that all monocytes and macrophages in these tumors are very high for CD11c, which is most likely not the case, and is contradicted by the CD11c/MHC-II plot.

- Response: The correction noted above also addresses this point. With the Axis corrected, the major population of cells is indeed CD11b⁺ CD11c⁻.

- The Mac gating (Ly6C/F4/80 plot) follows a pre-gating on cells that are mostly MHC-II-positive (in the SSC-A/MHC-II plot). Nevertheless, 34% of these Macs are MHC-II-negative, which is again a very surprising finding. In the SSC-A/MHC-II plot there are indeed also MHC-II-negative cells, but these have a very high SSC, suggesting that these cells are rather eosinophils than macrophages.

- Response: We agree with the referee and have modified Figure S1B accordingly.

(2) The authors mention that MoDC can associate with a better CD8⁺ T-cell response (ref 37). MoDC have also been reported to be mainly immunosuppressive (Laoui et al, Nat Commun, 2016, 7:13720), so this possibility should be included as well.

Response: We have modified the text and included the suggested reference (Pag. 9, ref. 37)

(3) I agree that the KO macrophages are overall more M1-oriented and thus likely more immunogenic. However, a combined upregulation of iNOS and arginase is often associated with the induction of an immunosuppressive program. This should at least be mentioned in the text.

Response: This has been mentioned in the text (Pag. 12)

Reviewers' Comments:

Reviewer #2:

Remarks to the Author:

The authors successfully addressed all concerns.